# Exploratory Memory-Augmented LLM Agent via Hybrid On- and Off-Policy Optimization

**Zeyuan Liu[1*], Jeonghye Kim[1,2*], Xufang Luo[1†], Dongsheng Li[1], Yuqing Yang[1]**
[1]Microsoft Research  [2]KAIST
gritmaybe@gmail.com, jeonghye.kim@kaist.ac.kr,
{xufluo, dongsli, yuqyang}@microsoft.com
🔗 project page  ⭘ agent-lightning/empo2

## Abstract

Exploration remains the key bottleneck for large language model agents trained with reinforcement learning. While prior methods exploit pretrained knowledge, they fail in environments requiring the discovery of novel states. We propose Exploratory Memory-Augmented On- and Off-Policy Optimization (EMPO[2]), a hybrid RL framework that leverages memory for exploration and combines on- and off-policy updates to make LLMs perform well with memory while also ensuring robustness without it. On ScienceWorld and WebShop, EMPO[2] achieves 128.6% and 11.3% improvements over GRPO, respectively. Moreover, in out-of-distribution tests, EMPO[2] demonstrates superior adaptability to new tasks, requiring only a few trials with memory and no parameter updates. These results highlight EMPO[2] as a promising framework for building more exploratory and generalizable LLM-based agents.

## 1 Introduction

Large Language Models (LLMs) have recently emerged as powerful agents capable of reasoning, planning, and interacting with external environments (Achiam et al., 2023; Park et al., 2023; Yao et al., 2023; Kim et al., 2025). When combined with reinforcement learning (RL), such agents can adapt their behavior based on experience and feedback, enabling them to go beyond static prompting or supervised fine-tuning (Guo et al., 2025; Tan et al., 2024). This paradigm has driven recent

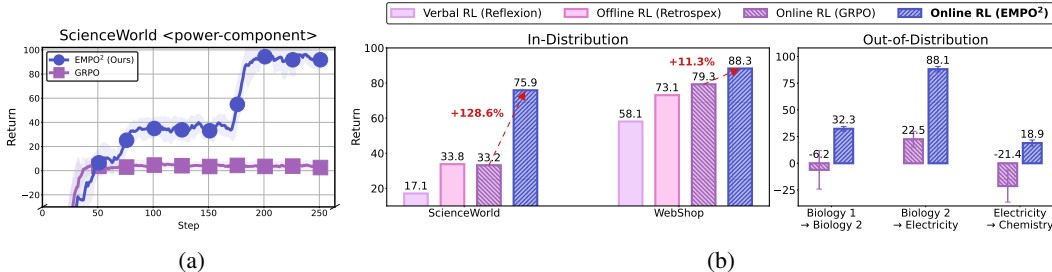

Figure 1: (a) **Comparison of the learning curves of GRPO and EMPO[2] (ours)** on the ScienceWorld `power-component` task. While GRPO converges to suboptimal performance, EMPO[2] continues to improve and accomplish the task. (b) **Comparison of EMPO[2] and other baselines in in-distribution (ID) and out-of-distribution (OOD) settings** on and WebShop. In ID experiments, it adapts well to familiar environments, achieving **128.6% on ScienceWorld** and **11.3% on Webshop** improvements over GRPO. In OOD experiments, it also shows strong performance with few trials and no weight updates, indicating effective use of memory to explore unfamiliar environments. Full results are in Tables 1, 2, and Figure 8.

---

* Equal contribution; work done during an internship at Microsoft Research.
† Corresponding author.

progress in areas such as interactive decision-making, tool use, and embodied AI (Feng et al., 2025b; Lu et al., 2025b; Feng et al., 2025a; Dong et al., 2025; Luo et al., 2025).

However, a key limitation of current LLM-based agents lies in their reliance on exploiting prior knowledge rather than engaging in systematic exploration. While RL frameworks emphasize balancing exploration and exploitation, many LLM-agent systems primarily leverage pretrained knowledge and conduct only limited search within familiar distributions. As a result, these agents often struggle in environments where progress depends on discovering novel states or actively acquiring new information, rather than reusing what is already known.

To address this challenge, recent research has incorporated external memory modules into LLMs as a form of long-term memory. This enables models to leverage past experiences to correct failed attempts, thereby improving decision-making in subsequent trials without requiring parameter updates (Shinn et al., 2023; Zhang et al., 2023). However, as noted in Zhang et al. (2023), the performance of such methods tends to saturate quickly, since collecting experiences with static parameters cannot fully capture the diversity needed for continuous improvement.

In this work, we present a unified framework that enables LLM agents to learn more effectively through broader exploration by jointly updating their **parametric policy parameters** with RL and their **non-parametric memory module** through interaction. Crucially, the non-parametric updates not only complement but also enhance the efficiency of parametric learning, thereby enabling more effective exploration and adaptation. This dual-update paradigm serves as a bridge between parameter-level optimization and memory-augmented reasoning. While memory is utilized during learning, moving toward more generalizable intelligence requires reducing dependence on external memory and instead embedding its benefits directly into the model's parameters. To this end, we pro-

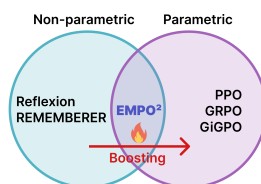

Figure 2: Non-parametric updates can encourage exploration, bootstrapping parametric updates.

pose **E**xploratory **M**emory-Augmented **On-** and **Off-P**olicy **O**ptimization (EMPO$^2$), a new hybrid RL algorithm that incorporates two modes in the rollout phase—depending on whether memory is used—and two modes in the update phase—on-policy and off-policy learning—thereby enabling agents to leverage memory when available while remaining robust in its absence.

In our experiments, we evaluate EMPO$^2$ on two widely used multi-step embodied reasoning environments that require exploration to solve complex tasks: ScienceWorld (Wang et al., 2022) and WebShop (Yao et al., 2022). We compare its performance against a range of non-parametric and parametric (offline and online) RL approaches. As summarized in Figure 1, EMPO$^2$ substantially outperforms prior algorithms, achieving a 128.6% improvement on ScienceWorld and an 11.3% improvement on WebShop over the strong online RL baseline GRPO. The training curve in Figure 1 (a) further shows that, unlike GRPO, which converges prematurely to a suboptimal solution, EMPO² leverages continuous exploration and successfully solves the task. Moreover, for the OOD experiments (Figure 1, rightmost), the model also achieves good scores with only a few trials and no weight updates, indicating that the updated model has acquired the ability to use memory to explore unseen or unfamiliar environments. These results highlight EMPO$^2$ as a promising direction for building more adaptive and generalizable embodied agents.

## 2 PRELIMINARIES

Online RL consists of alternating between a rollout phase, in which trajectories are generated using the current policy $\pi$ parameterized by $\theta$, and an update phase, in which the policy is optimized based on those rollouts.

**Policy Rollout.** We consider a setting where, given a sampled task $u \sim p(\mathcal{U})$, an LLM agent solves the task through multi-step interactions with the environment. Starting from task $u$, the LLM $\pi_\theta$ generates the first natural-language action $a_1 \sim \pi_\theta(\cdot \mid u) \in \mathcal{A}$. Executing this action, the environment returns a reward $r_1$ and the next state $s_1$. At a general timestep $t$, conditioned on the current state $s_t$ and the task $u$, the policy produces the next action $a_{t+1} \sim \pi_\theta(\cdot \mid s_t, u)$. This interaction loop continues until the task is completed or a maximum number of steps is reached. A rollout trajectory is thus defined as the sequence of states, actions, and rewards, $\tau = (u, a_1, r_1, s_1, a_2, r_2, \ldots, s_T)$.

**Group Relative Policy Optimization.** Group Relative Policy Optimization (GRPO) (Shao et al., 2024) updates the policy by comparing multiple rollouts of the same task $u$, removing the need for the value function in PPO (Schulman et al., 2017). Given a task $u$, the policy $\pi_\theta$ generates $N$ rollout trajectories $\{\tau^{(1)}, \ldots, \tau^{(N)}\}$. Each trajectory receives a return $\{R^{(1)}, \ldots, R^{(N)}\}$, defined as the sum of rewards along the trajectory: $R^{(i)} = \sum_{t=1}^{T} r_t^{(i)}$.. For each action $a_t^{(i)}$ taken in trajectory $\tau^{(i)}$, we define its relative advantage as: $A(a_t^{(i)}) = \frac{R^{(i)} - \frac{1}{N}\sum_{j=1}^{N} R^{(j)}}{\sigma(R)}$, where actions from trajectories with higher-than-average reward obtain positive advantage, while those from lower-performing ones obtain negative advantage. The GRPO loss is then:

$$\mathbb{E}_{\substack{u \sim p(\mathcal{U}) \\ \{\tau^{(i)}\}_{i=1}^{N} \sim \pi_{\theta_{\text{old}}}}} \left[ \frac{1}{NT} \sum_{i=1}^{N} \sum_{t=1}^{T} \min\left( \rho_\theta(a_t^{(i)}) A(a_t^{(i)}), \text{clip}\left(\rho_\theta(a_t^{(i)}), 1-\epsilon, 1+\epsilon\right) A(a_t^{(i)}) \right) \right]$$
$$- \beta\, D_{\text{KL}}\left( \pi_\theta(\cdot|u) \,\|\, \pi_{\text{ref}}(\cdot|u) \right), \tag{1}$$

where $\rho_\theta(a_t^{(i)}) = \frac{\pi_\theta(a_t^{(i)}|s_t^{(i)}, u)}{\pi_{\theta_{\text{old}}}(a_t^{(i)}|s_t^{(i)}, u)}$, with $\beta \geq 0$ controlling the regularization strength toward a reference policy $\pi_{\text{ref}}$.

## 3 THE EXPLORATION PROBLEM OF LLM AGENTS

LLMs encode rich prior knowledge, but such priors often fail to reflect the actual rules or dynamics of a given environment. Blind reliance on these priors can lead to erroneous behaviors, making it necessary for agents to adapt through direct interaction and trial-and-error. A key requirement for such adaptation is **exploration**, which involves seeking information beyond pre-training, sometimes by taking atypical or counterintuitive actions. However, current LLM-based agents struggle with this (Qiao et al., 2024; Zhou et al., 2024), as it demands stepping outside the distribution of behaviors where the model feels most confident.

Consequently, many prior studies have sought to align agents with new environments through warm-start supervised fine-tuning (SFT) using numerous golden trajectories (Song et al., 2024; Qiao et al., 2024; Xiang et al., 2024), leveraging large-scale models such as GPT-4 (Tang et al., 2024; Lin et al., 2023), or employing human engineering or well-established simulation information (Choudhury & Sodhi, 2025). While these methods achieve strong results in constrained settings, their effectiveness is limited to cases where such external support is available, and they generalize poorly to unseen scenarios without it.

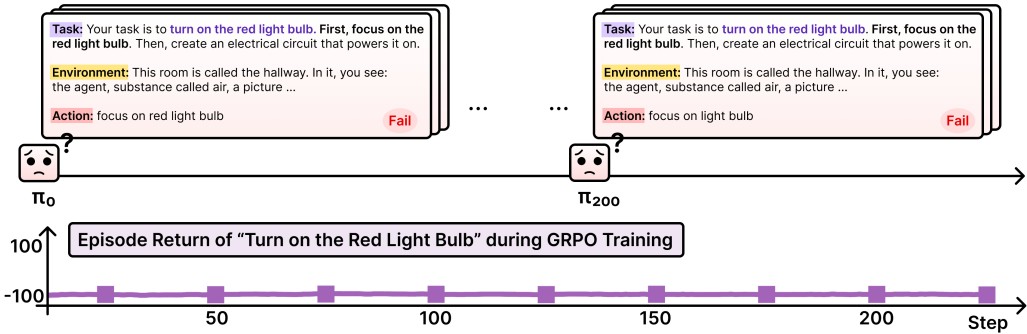

Figure 3: **When training LLM with GRPO in ScienceWorld, the agent struggles because of insufficient exploration.** For instance, in the task "turn on the red light bulb," the agent must first find the red light bulb before activating it. However, the agent fails to locate it and, as a result, cannot complete the task. Rather than analyzing the cause of failure and exploring alternative actions, the agent proceeds unchanged, so its score stagnates even as additional training steps are taken.

Therefore, we focus on how to efficiently train agents in online RL through trial and error, without any prior embedding of the environment's rules. The key challenge is that, without intrinsic exploration capability, online RL struggles to optimize effectively. As illustrated in Figure 3, in ScienceWorld (Wang et al., 2022) environment the agent is given the mission "turn on the red light

bulb." The instructions specify that the agent should first focus on the light bulb and then build a circuit to activate it, based on the current room observation. However, since no red light bulb is present in the observation, the agent must search the environment to locate it. Instead, the agent follows the instruction literally, attempts to focus on the red light bulb, and fails because it does not exist in the room. Ideally, when an agent fails to reach its goal, it should analyze the reasons for failure and broaden its action space to discover successful strategies. Yet in representative online RL algorithms GRPO (Shao et al., 2024), prior trajectory rollouts provide no continuity beyond a scalar reward signal, thereby restricting exploration and ultimately limiting learning.

## 4 METHOD

In this section, we present Exploratory Memory-augmented On- and Off-Policy Optimization (EMPO$^2$), a novel algorithm aimed at tackling the exploration challenges in online RL. EMPO$^2$ operates in two modes for both rollout phase and update phase. During rollout, actions can be generated either through **(1) prompting without memory**, where no retrieved information is used, or **(2) memory-augmented prompting**, conditioned on tips retrieved from memory. In the update phase, rollouts with memory-augmented prompting are used in two ways: **(a) on-policy**, where tips are retained and the update is performed with the original prompt, and **(b) off-policy**, where tips are removed during update. Notably, tips are generated not by a separate model but by the policy $\pi_\theta$ itself, which is continually updated during training. The full algorithm is provided in Appendix A.

### 4.1 ADVANCING EXPLORATION WITH SELF-GENERATED MEMORY

A key component of EMPO$^2$ is its use of memory to maintain continuity across rollouts. Information obtained from an agent's interactions can be encoded into parameters through policy optimization, but it can also be recorded in an external memory that the agent continuously consults. Since our policy is initialized from a pretrained LLM with inherent summarization and reflection abilities, these abilities can be leveraged as auxiliary signals in addition to scalar rewards, thereby guiding exploration more effectively. To realize this, EMPO$^2$ integrates both *parametric* (parameter updates within the LLM) and *non-parametric* (external memory) updates, strengthening the linkage between rollouts and promoting exploration, with all data and guidance generated autonomously by the agent.

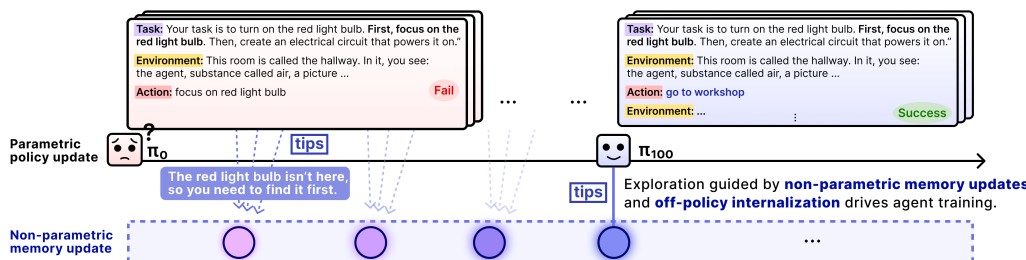

Figure 4: In EMPO$^2$, the current policy parameters $\pi_\theta$ are used to review past rollouts, with the resulting insights added to memory. This updated memory conditions subsequent rollouts and promotes exploration.

In the non-parametric updates, similar to Reflexion (Shinn et al., 2023), the agent reviews past rollouts, generates self-guidance tips, and stores them in memory. These tips help the agent avoid repeated mistakes and explore new strategies. Unlike Reflexion, focuses on iterative verbal guidance to achieve higher rewards in the next trial, our approach aims for these tips to lead to enhanced exploration that is ultimately consolidated through parametric updates.

**Self-Generated Memory and Tips.** We define a memory buffer $\mathcal{M} = \{\text{tip}_1, \text{tip}_2, \ldots\}$, which stores reflective tips generated by the policy $\pi_\theta$ during trajectory reflection. Formally, when an episode $i$ of task $u$ terminates at timestep $t$, the policy takes the final state $s_t$ together with a tip-generation prompt as input and produces a tip, where $\text{tip}_i \sim \pi_\theta(s_t, u, \text{tip-generation prompt})$. A set of illustrative examples is provided below, while the tip-generation prompt is presented in Appendix B, and additional examples are included in Appendix E.1.

> **Examples of Generated Tips – ScienceWorld <power-component> task**
>
> - You moved between the kitchen and bathroom but did not find a green wire or a green light bulb to connect.
> - You focused on the red light bulb but did not complete the task of turning on the red light bulb. You are in the hallway and need to find a way.
> - The trajectory involves connecting the battery to the green wire terminals to power the green light bulb, but the connections to air and other objects are irrelevant.
> - The circuit for the green light bulb was partially connected but still missing the battery connection; the task is not fully completed.

## 4.2 PARAMETERIZE NON-PARAMETRIC UPDATES VIA HYBRID POLICY OPTIMIZATION

Agents can use memory to improve exploration and learning efficiency, but the acquired knowledge needs be internalized into model parameters to enhance inherent capabilities. To this end, we propose two modes for the rollout and update phases, whose combinations yield three hybrid learning modes (Figure 5).

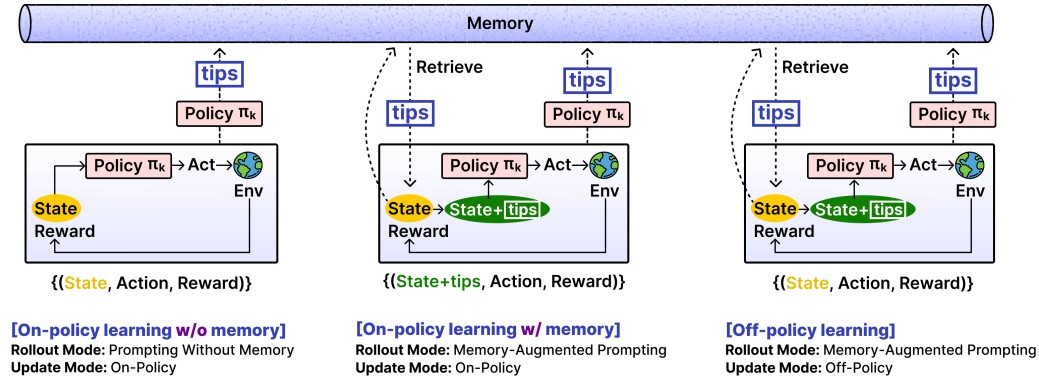

Figure 5: **EMPO$^2$ mode combinations.** By combining the two rollout modes and update modes, three EMPO mode configurations are possible: on-policy learning without memory, on-policy learning with memory and off-policy learning.

**Rollout Modes.** During rollouts, the agent samples between the two modes, selecting one mode at each step: mode (2) with memory rollout probability $p$ and mode (1) with probability $1 - p$. The ablation study of $p$ can be found in Appendix F.1.

(1) **Prompting Without Memory.** For each task $u$, at each timestep $t$, the policy $\pi_\theta$ generates actions conditioned only on the current state $s_t$ and the task $u$: $a_{t+1} \sim \pi_\theta(\cdot \mid s_t, u)$.

(2) **Memory-Augmented Prompting.** For each task $u$, at each timestep $t$, a retrieval operator $\mathrm{Retr}(o_t; \mathcal{M}) \subseteq \mathcal{M}$ selects tips most relevant to the current state $s_t$, e.g., via similarity search in the embedding space. We denote the retrieved set as $\boxed{\text{tips}}_t$. In memory-augmented prompting, the policy conditions its action on both $s_t$ and $\boxed{\text{tips}}_t$: $a_{t+1} \sim \pi_\theta(\cdot \mid s_t, u, \boxed{\text{tips}}_t)$. We limit the number of retrieved tips at 10.

**Update Modes.** Trajectories generated under rollout mode (1) are directly used for updates, whereas those generated under rollout mode (2)—memory-augmented prompting—follow one of two update modes chosen at random during the update phase. Mode (b) is selected with off-policy update probability $q$, and mode (a) with probability $1 - q$. The ablation study of $q$ can be found in Appendix F.1.

(a) **On-Policy Updates.** On-policy update uses the same prompt as in the rollout, and $\rho_\theta(a_t^{(i)})$ in eq.1 becomes $\rho_\theta(a_t^{(i)}) = \frac{\pi_\theta(a_t^{(i)} \mid s_t^{(i)}, u, \boxed{\text{tips}}_t)}{\pi_{\theta_{\text{old}}}(a_t^{(i)} \mid s_t^{(i)}, u, \boxed{\text{tips}}_t)}$.

(b) **Off-Policy Updates.** In this mode, the stored log-probabilities $\ell_t^{\text{tips}} = \log \pi_\theta(a_t \mid s_t, u, \boxed{\text{tips}}_t)$ are replaced with the log-probabilities assigned by the same policy $\pi_\theta$ when conditioned only on $(s_t, u)$, namely $\ell_t^{\text{no-tips}} = \log \pi_\theta(a_t \mid s_t, u)$. In this formulation, the advantage update is performed based on how natural the action appears under the distribution without tips.

This construction can be interpreted as a form of **reward-guided knowledge distillation**. Trajectories sampled under the tips-conditioned policy act as teacher demonstrations, while the student policy $\pi_\theta(\cdot \mid s, u)$ is updated to reproduce those trajectories in proportion to their advantage. High-reward trajectories ($\hat{A}_t > 0$) are reinforced, while low-reward trajectories ($\hat{A}_t < 0$) are suppressed, resulting in selective distillation that emphasizes beneficial behaviors. In this way, tips serve as an intermediate scaffolding mechanism that improves exploration and trajectory quality, while the reward signal ensures that only advantageous behaviors are ultimately retained. Consequently, the final policy learns to internalize the benefits of tip conditioning without requiring tips at inference time. Appendix C provides an illustrative breakdown and a summary table for the calculation of the importance sampling ratio.

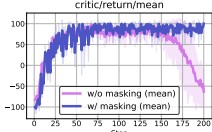

Figure 6: Masking tokens stabilizes training.

**Stabilizing Off-Policy Training.** Off-policy training is prone to instability and may collapse (see Figure 6). In such cases, gradient normalization, entropy loss, KL loss, and policy gradient loss can all diverge to NaN. Prior work, Yang et al. (2025) shows that low-probability tokens destabilize training by amplifying gradient magnitudes through unbounded likelihood ratios. Motivated by this, we introduce a masking mechanism that suppresses the advantage term for tokens whose probability under $\pi_\theta$ falls below a threshold $\delta$. Finally, the loss in Eq. 1 is modified as

$$\mathbb{E}_{\substack{u \sim p(\mathcal{U}) \\ \{\tau^{(i)}\} \sim \pi_{\theta_{\text{old}}}}} \left[ \frac{1}{NT} \sum_{i=1}^{N} \sum_{t=1}^{T} \min\left( \rho_\theta^{(i,t)} A(a_t^{(i)}), \ \text{clip}(\rho_\theta^{(i,t)}, 1-\epsilon, 1+\epsilon) A(a_t^{(i)}) \right) \cdot \mathbf{1}_{\pi_\theta(a_t^{(i)}|s_t^{(i)},u) \geq \delta} \right]$$
$$- \beta D_{\text{KL}}\left( \pi_\theta(\cdot|u) \,\|\, \pi_{\text{ref}}(\cdot|u) \right). \tag{2}$$

**Intrinsic Rewards for Exploration.** To further encourage exploration, and inspired by prior work on exploration-targeted online RL (Burda et al., 2018b; Bellemare et al., 2016; Ecoffet et al., 2019), we introduce an intrinsic reward based on the novelty of the current state. A memory list stores distinct states, and for each new state we compute its cosine similarity with existing entries. If the similarity falls below a threshold, the state is added to memory and assigned a reward. The intrinsic reward is defined as $r_{\text{intrinsic}} = \frac{1}{n}$, where $n$ denotes the number of similar past states. This mechanism encourages the agent to explore novel states even when no extrinsic reward is provided by the environment and maintains policy entropy, as shown in Figure 7.

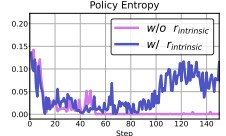

Figure 7: Policy entropy comparison with vs. without intrinsic rewards.

## 5 RELATED WORK

**LLM Agents in Multi-Step Embodied Tasks.** LLM agents for multi-step embodied tasks have been studied under different paradigms. Data-driven approaches (Song et al., 2024; Xiong et al., 2024; Qiao et al., 2025; 2024; Tajwar et al., 2025) enhance decision-making through effective data collection methods and imitation learning. Model-based agents (Tang et al., 2024; Zhou et al., 2024) build world models, often by generating code with large closed-source systems such as GPT-4. Other methods (Lin et al., 2023; Choudhury & Sodhi, 2025) strengthen reasoning through model transitions or by leveraging privileged information provided by the simulation environment. In contrast, our approach reduces reliance on such external resources and emphasizes autonomous growth through the agent's own exploration and self-improvement.

**Memory for LLM Agents.** To enable progressive improvement from past experiences, Reflexion (Shinn et al., 2023) and REMEMBERER (Zhang et al., 2023) leverage external memory. Reflexion stores verbal reflections for later prompting, while REMEMBERER records observations, actions, rewards, and Q-values, retrieving similar cases as few-shot exemplars. These methods show that LLMs can improve without parameter updates. However, with fixed parameters, they cannot expand intrinsic knowledge, so adaptation remains short-term (Zhang et al., 2023), relying on external memory rather than achieving long-term evolution and generalization.

**Learning by Knowledge Distillation** Our hybrid off-policy update functions as reward-guided knowledge distillation during online training. Snell et al. (2022) introduced context distillation, where the model first solves tasks using a Teacher prompt (with instructions, examples, explanations, and scratch-pad reasoning) and then learns to produce the final answer from a minimal Student prompt via offline, SFT-based distillation. In contrast, we integrate knowledge distillation into online RL, leveraging online adaptability while enhancing exploration for more efficient training.

**RL for LLM Agents.** RL provides a robust framework for optimizing LLM parameters through observations and reward signals from environment interactions. Prior work, Retrospex (Xiang et al., 2024), showed that offline RL, which learns optimal policies from large logged datasets, can improve LLM agent performance. Recent studies focus on online RL (Shao et al., 2024; Feng et al., 2025b; Wang et al., 2025), where agents learn in real time. GiGPO (Feng et al., 2025b) advanced GRPO by grouping rollouts with similar observations, enabling finer credit assignment and stronger performance. Our work advances this online RL direction by integrating non-parametric memory updates into both on- and off-policy learning, yielding substantially higher sample efficiency.

**Enhancing Exploration for Online RL.** A central challenge in online RL is effective exploration. Classical methods such as count-based exploration (Bellemare et al., 2016) and Random Network Distillation (Burda et al., 2018b) use intrinsic rewards to encourage novelty. Go-Explore (Ecoffet et al., 2019) stores key states and re-explores from them, solving hard-exploration tasks like Atari games. Its LLM extension, Intelligent Go-Explore (Lu et al., 2025a), achieves strong results in environments such as TextWorld (Côté et al., 2018), but relies on large closed-source models and does not perform parameter updates. In our concurrent work, RLVMR (Zhang et al., 2025) employs warm-start SFT to elicit diverse reasoning types (planning, exploration, and reflection) and provides dense, process-level rewards for each reasoning type during online RL, enhancing exploration and credit assignment. Together, these studies underscore the importance of structured exploration for scaling RL to complex environments.

## 6 EXPERIMENTS

To examine the effectiveness of $EMPO^2$, we conduct extensive experiments on two widely used LLM agent benchmarks: ScienceWorld (Wang et al., 2022) and WebShop (Yao et al., 2022) using Qwen2.5-7B-Instruct (Qwen et al., 2025) as the base model. The $EMPO^2$ performance we evaluate is the performance of the trained model without memory at test time.

### 6.1 SCIENCEWORLD

ScienceWorld (Wang et al., 2022) is an interactive text-based benchmark in which an agent performs science experiments at the elementary school level. Successfully completing these experiments requires long-term multi-step planning, hypothesis testing, and interpretation of outcomes, as well as sufficient exploration to determine where the necessary tools are and what appropriate actions should be taken. ScienceWorld includes tasks from diverse topics and in our experiments, we cover 19 tasks spanning chemistry, classification, biology, electricity, and measurement.

**Baselines.** We compare $EMPO^2$ with several RL approaches. For non-parametric RL, Reflexion (Shinn et al., 2023) updates memory in a non-parametric manner by incorporating LLM reflections from previous trajectories and using them in the prompt for the subsequent trial. For offline RL, Retrospex (Xiang et al., 2024) leverages an SFT-trained model and uses a Q-function learned via Implicit Q-learning (Kostrikov et al., 2022) to dynamically rescore actions. The official Retrospex paper used the smaller Flan-T5-Large (Chung et al., 2024) (770M) and incorporated human-designed heuristics to assist the agent during evaluation. In contrast, to ensure consistency in our experimental setup, we standardize the base model of Retrospex to Qwen2.5-7B-Instruct and exclude these heuristics. Finally, for online RL, we include GRPO (Shao et al., 2024) as a representative baseline. Further details are provided in Appendix D.

**Training Details.** Our $EMPO^2$ implementation is based on verl (Sheng et al., 2024), one of the representative RL-for-LLM libraries. We extended GRPO in verl from a single-step setup to a multi-step setup and incorporated both a memory module and an off-policy loss calculation component. We use the same hyperparameter configuration for GRPO and $EMPO^2$. The prompt used is provided in Appendix B, and implementation details are given in Appendix D.2.

Table 1: **Comparison results of ScienceWorld.** Each task in ScienceWorld contains multiple variants. We use the first five variants for training and evaluate on the 20 unseen test variants. Bold shows the best performance per task, while red shading marks cases where parametric updates score lower than non-parametric updates. The EMPO$^2$ performance we evaluate is the performance of the trained model without memory at test time.

| Qwen2.5-7B-Instruct | | Naive | Non-Parametric | Offline RL | Online RL | |
|---|---|---|---|---|---|---|
| Topic | Task | | Reflexion | Retrospex | GRPO | EMPO$^2$ |
| Chem istry | chemistry-mix | $-42.0\pm38.0$ | $1.2\pm0.7$ | $20.8\pm10.0$ | $12.4\pm3.5$ | $\textbf{42.7}\pm\textbf{12.4}$ |
| | chemistry-mix-paint-secondary-color | $-33.0\pm47.1$ | $0.0\pm0.0$ | $27.8\pm6.3$ | $7.1\pm2.8$ | $\textbf{33.3}\pm\textbf{0.6}$ |
| | chemistry-mix-paint-tertiary-color | $-33.9\pm44.3$ | $36.9\pm5.7$ | $7.6\pm4.2$ | $\textbf{42.6}\pm\textbf{6.2}$ | $39.2\pm8.7$ |
| Classi fication | find-animal | $-58.2\pm50.2$ | $39.5\pm5.8$ | $25.9\pm13.5$ | $72.4\pm6.8$ | $\textbf{100.0}\pm\textbf{0.0}$ |
| | find-living-thing | $-65.1\pm48.1$ | $36.6\pm6.1$ | $20.6\pm4.8$ | $68.7\pm7.1$ | $\textbf{100.0}\pm\textbf{0.0}$ |
| | find-non-living-thing | $-35.9\pm68.6$ | $4.8\pm2.0$ | $89.1\pm11.5$ | $24.7\pm6.4$ | $\textbf{100.0}\pm\textbf{0.0}$ |
| | find-plant | $-47.1\pm66.2$ | $15.1\pm3.8$ | $23.0\pm3.5$ | $46.2\pm7.9$ | $\textbf{100.0}\pm\textbf{0.0}$ |
| Bio logy1 | identify-life-stages-1 | $-48.9\pm65.4$ | $9.2\pm2.4$ | $19.0\pm25.7$ | $17.9\pm4.7$ | $\textbf{36.2}\pm\textbf{11.2}$ |
| | identify-life-stages-2 | $-50.7\pm65.0$ | $33.8\pm5.5$ | $11.0\pm1.7$ | $39.5\pm6.0$ | $\textbf{56.3}\pm\textbf{8.1}$ |
| Bio logy2 | lifespan-longest-lived | $-51.8\pm64.8$ | $44.6\pm6.5$ | $55.0\pm15.0$ | $78.2\pm7.3$ | $\textbf{100.0}\pm\textbf{0.0}$ |
| | lifespan-longest/shortest-lived | $-56.2\pm63.5$ | $34.1\pm5.1$ | $38.0\pm15.0$ | $62.3\pm6.9$ | $\textbf{100.0}\pm\textbf{0.0}$ |
| | life-span-shortest-lived | $-56.8\pm63.0$ | $6.1\pm1.9$ | $67.0\pm23.8$ | $20.6\pm4.4$ | $\textbf{100.0}\pm\textbf{0.0}$ |
| Elec tricity | power-component | $-90.0\pm39.4$ | $6.3\pm1.8$ | $8.2\pm2.4$ | $15.1\pm3.9$ | $\textbf{94.3}\pm\textbf{3.6}$ |
| | power-component-renewable-vs-nonrenewable-energy | $-85.0\pm49.8$ | $11.7\pm2.9$ | $10.0\pm3.2$ | $24.6\pm5.5$ | $\textbf{92.6}\pm\textbf{0.9}$ |
| | test-conductivity | $-86.9\pm42.4$ | $13.2\pm3.1$ | $60.0\pm0.0$ | $27.8\pm6.1$ | $\textbf{89.5}\pm\textbf{3.2}$ |
| | test-conductivity-of-unknown-sub | $-81.7\pm48.6$ | $2.6\pm1.0$ | $65.5\pm23.7$ | $9.5\pm3.4$ | $\textbf{71.4}\pm\textbf{6.3}$ |
| Measu rement | measure-melting-point-known-sub | $-97.5\pm7.5$ | $11.4\pm3.0$ | $26.5\pm16.1$ | $19.8\pm5.0$ | $\textbf{27.6}\pm\textbf{4.2}$ |
| | use-thermometer | $-83.7\pm43.6$ | $0.9\pm0.4$ | $32.5\pm32.1$ | $7.6\pm2.5$ | $\textbf{82.7}\pm\textbf{13.3}$ |
| | **Average** | -61.3 | 17.1 | 33.8 | 33.2 | **75.9** |

**Main Results.**    Table 1 presents the comparison results among baselines. In ScienceWorld, failed tasks lead to negative rewards, producing returns between -100 and 100. The baseline Qwen2.5-7B-Instruct obtains an average return of -61.3, which improves to 17.1 when non-parametric RL (Reflexion) is applied. Offline RL (Retrospex) produces substantial performance gains compared to them, but in some tasks underperforms compared to non-parametric RL (highlighted in red). Online RL with GRPO also achieves considerable improvements, and its average performance is comparable to that of offline RL. However, unlike offline RL, it never underperforms non-parametric RL, indicating that online RL generalizes better to unseen variants. Our EMPO$^2$ demonstrates substantially higher learning performance compared to all baselines. Among the tasks that initially started with negative rewards, seven reached the maximum score of 100. On average, EMPO$^2$ achieved more than twice the performance improvement over GRPO, demonstrating its effectiveness in greatly enhancing learning efficiency in online RL.

**Adaptation in New Tasks with Memory Updates.**    An agent post-trained on a single task may exhibit limited ability to generalize to new scenarios. However, EMPO$^2$, which acquires the ability to explore by leveraging memory, demonstrates significantly stronger adaptability to novel situations compared to GRPO, which is trained without learning to utilize memory. Figure 8 illustrates how a model trained on one task adapts when memory is introduced in a new task. In particular, we demonstrate cases with varying levels of topic difference. For a relatively similar transition, we examine Biology 1 (*identify-life-stages-2*) → Biology 2 (*life-span-shortest-lived*). For a more distinct transition, we examine Biology 2 (*lifespan-longest-lived*) → Electricity (*test-conductivity*), and Electricity (*power-component*) → Chemistry (*chemistry-mix-paint-secondary-color*).

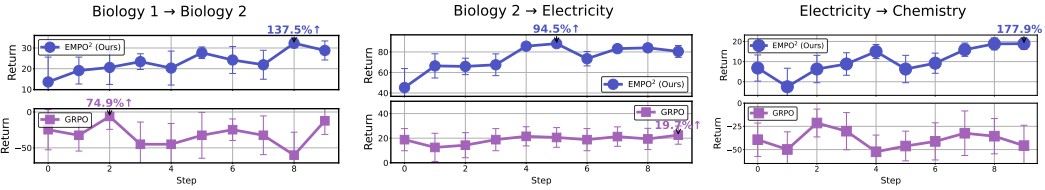

Figure 8: Comparison of GRPO and EMPO$^2$ adapting to new tasks. Step 0 has no memory, while later steps use accumulated memory as in EMPO$^2$ training.

As shown in Figure 8, without memory (step 0), EMPO$^2$ achieves stronger baseline performance on novel tasks than GRPO. When memory is enabled, EMPO$^2$ adapts rapidly, yielding an average improvement of 136% across three scenarios within 10 steps. GRPO, by contrast, demonstrates notable gains in some cases but exhibits greater variability and, in other instances, fails to adapt to unfamiliar tasks. In certain situations, its performance even falls below that of the Qwen2.5-7B-Instruct base model. Though these findings are preliminary, they indicate that EMPO$^2$ has strong potential as an RL framework for developing agents that are both more general and adaptable.

## 6.2 WEBSHOP

WebShop (Yao et al., 2022) is an HTML-based online shopping environment where agents search, navigate, and purchase items according to user instructions. When the "buy" action is selected, a final reward is given based on how well the product's attributes and price match the criteria.

**Baselines.** For the WebShop experiments, we use the same baselines as in the ScienceWorld experiments, with the addition of one more online RL baseline, GiGPO (Feng et al., 2025b), as GiGPO does not cover ScienceWorld but provides benchmarking results on WebShop. The scores of Naive, Reflexion, GRPO, and GiGPO are taken from Feng et al. (2025b), while Retrospex results are re-run using the official Retrospex code with the Qwen2.5-7B-Instruct model.

**Training Details.** The WebShop EMPO$^2$ implementation builds on the official GiGPO (Feng et al., 2025b) code with the same hyperparameters. Further details are provided in Appendix D.3.

**Main Results.** Table 2 presents the baseline comparison results on WebShop. Consistent with the findings in ScienceWorld, EMPO$^2$ once again delivers the strongest performance. Although offline RL, online GRPO, and GiGPO each outperform non-parametric RL, GiGPO further enhances GRPO by leveraging additional advantage estimation through grouping similar observations within rollout groups. Despite these gains, EMPO$^2$ surpasses all baselines, achieving both higher scores and success rates than GiGPO. Taken together, these results indicate that EMPO$^2$ consistently demonstrates superior performance in web-based environments due to its improved exploration.

Table 2: **Comparison results of WebShop.** Following Feng et al. (2025b), we average results over three random seeds and report both the mean score and the mean success rate (%). GiGPO$_{w/\ std}$ denotes the use of the normalization factor $F_{norm} = std$, whereas GiGPO$_{w/o\ std}$ uses $F_{norm} = 1$, as specified in Feng et al. (2025b). The EMPO$^2$ performance we evaluate is the performance of the trained model without memory at test time.

| Qwen2.5-7B-Instruct | Naive | Non-Parametric | Offline RL | | Online RL | | |
| --- | --- | --- | --- | --- | --- | --- | --- |
| | | Reflexion | Retrospex | GRPO | GiGPO w/ std | GiGPO w/o std | **EMPO$^2$** |
| Score | 26.4 | 58.1 | 73.1±4.1 | 79.3±2.8 | 84.4±2.9 | 86.2±2.6 | **88.3±2.6** |
| Succ. | 7.8 | 28.8 | 60.4±3.9 | 66.1±3.7 | 72.8±3.2 | 75.2±3.8 | **76.9±4.1** |

## 6.3 ABLATION STUDY ON MODE COMBINATIONS

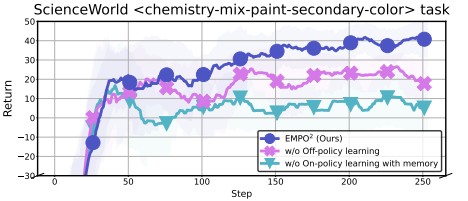 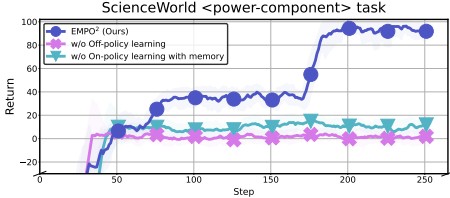

Figure 9: Comparison of training curves between EMPO$^2$ and variants that exclude either off-policy learning or on-policy learning with memory.

As shown in Figure 5, EMPO$^2$ incorporates three mode combinations: on-policy learning without memory, on-policy learning with memory, and off-policy learning, and we further analyze how leveraging each affects performance on two ScienceWorld tasks, where EMPO$^2$ shows significant improvements over GRPO. Figure 9 presents training curves comparing EMPO$^2$ with variants that exclude either off-policy or on-policy learning with memory. As shown in the graphs, removing

either component results in suboptimal learning, indicating that a balanced integration of on-policy and off-policy updates is most effective for performance improvement. This highlights their complementary roles: on-policy updates contribute to stable learning, while off-policy updates enable reasoning as if guided by additional tips, and their combination yields both faster convergence and stronger final performance.

# 7 CONCLUSION

In this work, we propose EMPO$^2$, a novel RL method that enhances exploration in parametric RL by leveraging non-parametric memory updates. EMPO$^2$ integrates both on-policy and off-policy learning, thereby improving training efficiency and stability. Our experiments demonstrate that EMPO$^2$ achieves remarkable gains in training efficiency on ScienceWorld and WebShop, and further shows the ability to adapt rapidly to new domains in a few-shot manner by incorporating additional memory. An ablation study confirms the importance of the three distinct modes of EMPO$^2$.

While our study demonstrates the potential of EMPO$^2$ as a RL framework for general agents, our current implementation for memory employs a simple similarity-based search for memory retrieval. More advanced retrieval mechanisms may further enhance performance. Moreover, although our experiments primarily utilize Qwen2.5-7B-Instruct, extending EMPO$^2$ to a broader range of model families and sizes could yield deeper insights into its generality and robustness. In particular, scaling to larger models may further amplify the benefits of our approach. Beyond model scaling, applying EMPO$^2$ to new domains such as mathematics, coding, multi-hop question answering, and multimodal RL represents an exciting and challenging direction for future research. In addition, exploring other off-policy techniques beyond importance sampling could be of interest to achieve more stable and efficient hybrid optimization.

## ETHICS STATEMENT

This work evaluates EMPO$^2$ on ScienceWorld and WebShop, which are publicly available research benchmarks that do not include private data or sensitive information. We complied with dataset licenses and community standards for responsible use and citation, and no additional data collection or modification of the environments was performed.

Although our method exhibits strong adaptability in exploration and reasoning tasks, online RL systems may be misapplied in safety-critical real-world contexts. To reduce such risks, we confine our study to benchmark environments, and for real-world applications, responses generated by LLMs will require more careful scrutiny. We hope that future research will further address safety and broader societal impacts when extending embodied reasoning agents beyond simulation.

## REPRODUCIBILITY STATEMENT

We release an Agent Lightning (Luo et al., 2025) version of EMPO$^2$ at `agent-lightning/empo2`. Additionally, we provide detailed training information, including pseudocode in Appendix A, the hyperparameters used in our experiments, the hyperparameters for the baseline experiments, the GPU resources utilized, and code snippets for the additional components implemented in Appendix D.

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

## A  PSEUDO CODE

Algorithm 1 presents the pseudocode of EMPO$^2$. Compared to the original GRPO algorithm, EMPO$^2$ introduces several new components: a memory buffer, and tip retrieval and addition, and two rollout modes and two update modes.

---

**Algorithm 1** EMPO$^2$: Exploratory Memory-Augmented On- and Off-Policy Optimization

---

1: **Inputs:** Initial policy $\pi_{\theta_{\text{old}}}$, memory buffer $\mathcal{M}$, task distribution $p(\mathcal{U})$, group size $N$, batch size $B$, max episode length $T$
2: **for** each training iteration **do**
3:     {// **Multi-step rollout**}
4:     Sample $B$ tasks $u \sim p(\mathcal{U})$ and initialize $N$ identical environments (total $B \times N$)
5:     Sample $m_{\text{rollout}} \sim \{$Prompting Without Memory $: p,$ Memory-Augmented Prompting $: 1-p\}$
6:     Initialize state $s_0^{(i)} \leftarrow u^{(i)}$ for $i = 0, \ldots, B \times N - 1$
7:     **for** $t = 0$ to $T - 1$ **do**
8:       **for** $i = 0$ to $B \times N - 1$ **do**
9:         **if** $m_{\text{rollout}} =$ Memory-Augmented Prompting **then**
10:           $\boxed{\text{tips}}_t \leftarrow \text{Retr}(s_t^{(i)}; \mathcal{M})$
11:           Sample $a_t^{(i)} \sim \pi_\theta^{\text{old}}(\cdot \mid s_t^{(i)}, \boxed{\text{tips}}_t, u^{(i)})$
12:         **else**
13:           Sample $a_t^{(i)} \sim \pi_\theta^{\text{old}}(\cdot \mid s_t^{(i)}, u^{(i)})$
14:         **end if**
15:         Execute $a_t^{(i)}$, observe $r_t^{(i)}, s_{t+1}^{(i)}$
16:       **end for**
17:     **end for**
18:     **for** $i = 0$ to $B \times N - 1$ **do**
19:       Sample $\boxed{\text{tips}} \sim \pi_\theta^{\text{old}}(\cdot \mid s^{(i)}, u^{(i)}, \text{tip-generation prompt})$
20:       Append $\boxed{\text{tips}}$ to $\mathcal{M}$
21:     **end for**
22:     {// **Policy update**}
23:     **if** $m_{\text{rollout}} =$ Memory-Augmented Prompting **then**
24:       Sample $m_{\text{update}} \sim \{$On-Policy $: q,$ Off-Policy $: 1-q\}$
25:       **if** $m_{\text{update}} =$ Off-Policy **then**
26:         **for** $i = 0$ to $B \times N - 1$ **do**
27:           $\log \pi_{\theta_{\text{old}}}(a \mid s_t^{(i)}, \boxed{\text{tips}}_t, u^{(i)}) \leftarrow \log \pi_{\theta_{\text{old}}}(a \mid s_t^{(i)}, u^{(i)})$
28:         **end for**
29:       **end if**
30:     **end if**
31:     Update policy $\theta$ using the loss function in Eq. 2.
32: **end for**

---

## B  PROMPTS

The following prompts were used in our experiments. The ScienceWorld and WebShop prompts were used identically for both the online RL baseline and EMPO$^2$, with the WebShop prompt adapted from Feng et al. (2025b). The content inside the curly brackets ({}) is dynamically filled based on the current progress at each episode step.

---

**Tip Generation Prompt**

Thanks for your playing. Now you have ended a trajectory and collect some meaningless or valuable information from the interactions with the environment. Please summary the trajectory, and also summary what information you get from this trajectory, and how far this trajectory is from fully completing the task. Please response with only one sentence with only one line, do not include any extra words. You sentence should be less than 100 words.

---

---

**Prompt for ScienceWorld**

You have done a few science experiments successfully and below are the action history of your experiments with similar tasks. Here is 2 examples: {example_1} {example_2} Follow the report of the two example tasks shown to you previously, try to solve a similar new task.

Task Description: {task_description}
All your possible action formats are: {available_action_description}

If you enter an unfamiliar room for the first time, you can use the action 'look around' to discover the objects in it. Items in your inventory: {inventory}

Important! You can only use FOCUS actions on these items: {focus_items}. You cannot FOCUS on any other things. Please only use FOCUS as required by the task description. Also, please FOCUS more directly, try not to focus on the container. You could try to explore different actions, especially when you are not sure what the best action for your current observation.
{current_observation}

---

**Prompt for WebShop**

You are an expert autonomous agent operating in the WebShop e-commerce environment.

Your task is to: {task_description}.

Prior to this step, you have already taken {step_count} step(s). Below are the most recent {history_length} observations and the corresponding actions you took: {action_history}
You are now at step current_step and your current observation is: {current_observation}.

Your admissible actions of the current situation are:
[{available_actions}].

Now it's your turn to take one action for the current step.

You should first reason step-by-step about the current situation, then think carefully which admissible action best advances the shopping goal. This reasoning process MUST be enclosed within <think> </think> tags.

Once you've finished your reasoning, you should choose an admissible action for current step and present it within <action> </action> tags.

---

## C   DETAILED EXPLANATION OF IMPORTANCE SAMPLING RATIOS IN POLICY UPDATES

To further clarify our policy update mechanism, this section details the calculation of the importance sampling ratio $\rho_\theta$. The specific calculation depends on whether tips were used during the rollout and update phases. This leads to three distinct scenarios, as summarized in Table 3. The importance sampling ratio $\rho_\theta$ is defined as the ratio of the probability of an action under the current policy $\pi_\theta$ to its probability under the old policy $\pi_{\theta_{\text{old}}}$, used to correct for the distributional shift in off-policy learning.

Table 3: Calculation of the importance sampling ratio $\rho_\theta$ for different policy update modes. The ratio is computed as $\rho_\theta = \frac{\pi_\theta(a_t|\cdot)}{\pi_{\theta_{\text{old}}}(a_t|\cdot)}$, which in practice is often calculated using log-probabilities.

| Update Mode | Rollout Condition | Current Log Prob | Old Log Prob |
|---|---|---|---|
| Regular On-Policy | No tips | $\log \pi_\theta(a_t \mid s_t, u)$ | $\log \pi_{\theta_{\text{old}}}(a_t \mid s_t, u)$ |
| On-Policy w/ Tips | With $\text{tips}_t$ | $\log \pi_\theta(a_t \mid s_t, u, \text{tips}_t)$ | $\log \pi_{\theta_{\text{old}}}(a_t \mid s_t, u, \text{tips}_t)$ |
| Off-Policy | With $\text{tips}_t$ (rollout) | $\log \pi_\theta(a_t \mid s_t, u)$ | $\log \pi_{\theta_{\text{old}}}(a_t \mid s_t, u, \text{tips}_t)$ |

The three update modes shown in the table cover all scenarios. An update is considered on-policy when the policy used to generate actions ($\pi_{\theta_{old}}$) and the policy being updated ($\pi_\theta$) are conditioned on the same information. This applies to the first two modes:

- **Regular On-Policy**: This is the standard on-policy update. The conditioning context for both the current and old policies is identical $(s_t, u)$, with no tips involved.
- **On-Policy w/ Tips**: This mode is also on-policy because both policies are consistently conditioned on the provided tips $(s_t, u, \boxed{\text{tips}}_t)$.

The **Off-Policy** update is the key mechanism through which the model learns from external guidance. In this scenario, actions are sampled from the old policy augmented with tip information: $\pi_{\theta_{old}}(\cdot \mid s_t, u, \boxed{\text{tips}}_t)$. However, to "internalize" this guidance, the current log-probabilities for the new policy $\pi_\theta$ are recomputed without the tips, using only $\pi_\theta(a_t \mid s_t, u)$. This mismatch in conditioning between the new and old policies makes the update off-policy and allows the base policy to absorb the knowledge contained in the tips.

While the importance sampling variant in Table 3 is theoretically unbiased, the distribution shift can still lead to instability in practice. Therefore, this allows for different implementation choices, such as tuning the clipping scheme or computing the old log-probabilities without tips, in order to better control the bias–variance trade-off.

## D    EXPERIMENTS DETAILS

### D.1    RETROSPEX

In Retrospex (Xiang et al., 2024), the base models differ by environment: Flan-T5-Large (Chung et al., 2024) is used for ScienceWorld, while Llama-3-8B-Instruct (Grattafiori et al., 2024) is used for WebShop. To ensure consistency in our experiments, we standardized the base model to Qwen2.5-7B-Instruct. For this purpose, we utilized the offline trajectories provided by Retrospex and conducted SFT with LLaMA-Factory (Zheng et al., 2024). For the IQL (Kostrikov et al., 2022) Q-function, we employed the model released by Retrospex. During SFT training, we tuned the hyperparameters over learning rates $1.0 \times 10^{-5}, 5.0 \times 10^{-5}, 1.0 \times 10^{-6}$ and epochs $3, 8$, and adopted the configuration that yielded the best performance. Each run was conducted using two NVIDIA A100 GPUs with 80GB memory.

Moreover, in our Retrospex ScienceWorld evaluation, we remove human-designed heuristics to reduce reliance on manual rules. Retrospex normally skips any "focus on" action unless repeated three times or explicitly mentioned in the task, and replaces step-by-step "go to" actions with direct "teleport" moves. Removing these heuristics ensures the evaluation better reflects the agent's inherent capabilities.

### D.2    ONLINE RL: SCIENCEWORLD

We base our EMPO[2] implementation on the GRPO framework provided in verl (Sheng et al., 2024), while introducing the following key modifications:

- **Multi-step implementation:** In the original GRPO implementation in verl, an LLM rollout terminates after generating a single response to a given problem. We extend this to a multi-step setting, where the agent continues interacting with the environment until either a maximum episode length is reached or the environment issues a termination signal. This modification allows the agent to perform sequential reasoning and adapt its responses across turns.
- **Memory buffer integration:** To support EMPO[2]'s memory-based mechanism, we incorporate an explicit memory buffer. During multi-step rollouts, the agent can retrieve tips from memory and append newly generated tips to it. The code snippet for this part is as follows:

```python
import numpy as np, requests, uvicorn
from fastapi import FastAPI
from pydantic import BaseModel
from typing import List, Optional

app = FastAPI()
```

```python
cnt, mem_list, content_set = {}, {}, {}

class MemRequest(BaseModel):
    key: List[float]
    idx: Optional[int] = None
    content: Optional[str] = None
    score: Optional[float] = None

@app.post("/mem/")
async def mem_handler(req: MemRequest):
    global cnt, mem_list, content_set
    key, idx, content, score = req.key, req.idx, req.content, req.score

    if content == "Reset":  # Reset all buffers
        content_set = {i: set() for i in range(idx)}
        mem_list = {i: [] for i in range(idx)}
        cnt = {i: 0 for i in range(idx)}
        return {"status": "reset"}

    if content is not None:  # Store new memory
        if content not in content_set[idx]:
            content_set[idx].add(content)
            mem_list[idx].append(
                {"cnt": cnt[idx], "key": key, "content": content, "score": score})
            cnt[idx] += 1
            if len(mem_list[idx]) > 1000:  # Evict oldest
                content_set[idx].discard(mem_list[idx][0]["content"])
                mem_list[idx] = mem_list[idx][-1000:]
        return {"status": "added", "total": cnt[idx]}

    # Retrieve by cosine similarity (threshold > 0.5), return top-10 by score
    key_vec = np.array(key)
    candidates = []
    for m in mem_list[idx]:
        m_vec = np.array(m["key"])
        sim = np.dot(key_vec, m_vec) / (np.linalg.norm(key_vec) * np.linalg.norm(m_vec))
        if sim > 0.5:
            candidates.append(m)
    data = [x["content"] for x in sorted(candidates, key=lambda x: -x["score"])[:10]]
    return {"count": len(data), "data": data}

# --- Client utilities ---
def compress_text(text):
    return requests.post("http://127.0.0.1:8000/key_cal/", json={"text": text}).json()["key"
        ]

def retrieve_memory(idx, key):
    r = requests.post("http://127.0.0.1:8001/mem/", json={"key": key, "idx": idx}).json()
    return r["count"], r["data"]

def add_memory(idx, key, content, score):
    requests.post("http://127.0.0.1:8001/mem/",
                  json={"key": key, "idx": idx, "content": content, "score": score})

# --- Training-phase memory retrieval ---
if phase in ["on-policy-with-memory", "off-policy"]:
    text = "\n".join(f"{c['role']}:_{c['content']}" for c in conversations)
    key = np.array(compress_text(text)).reshape(-1).tolist()
    count, memories = retrieve_memory(buffer_id, key)
else:
    count, memories = 0, []

if __name__ == "__main__":
    uvicorn.run(app, host="0.0.0.0", port=8001, workers=4)
```

Listing 1: Implementation of memory buffer integration.

**Hyperparameters.** All online RL algorithms (GRPO, EMPO[2]) use the same hyperparameter configuration. The maximum response length is set to 32 tokens per step and 4,500 tokens in total, and each episode is limited to 30 steps. The actor learning rate is set to $1 \times 10^{-6}$. For GRPO, the group size is fixed at 8 and the mini-batch size at 16. The KL-divergence loss coefficient is set to 0.0. In addition, the actor rollout parameters are specified as follows: the clipping upper bound is set to 0.30, the clipping lower bound to 0.20, and the clipping ratio coefficient to 10.0.

**Computing Resources.** All experiments were conducted using eight NVIDIA A100 40GB GPUs.

## D.3 ONLINE RL: WEBSHOP

We base our EMPO$^2$ implementation on the GRPO framework provided in verl-agent (Feng et al., 2025b), and the modifications for EMPO$^2$ are the same as those described in Appendix D.2.

**Hyperparameters.** All online RL algorithms (GRPO, GiGPO, EMPO$^2$) use the same hyperparameter configuration, following Feng et al. (2025b). The maximum response length is set to 512 tokens, and each episode is limited to 15 steps. The actor learning rate is configured as $1 \times 10^{-6}$. For GRPO, the group size is fixed at 8. The rollout temperature is set to 1.0, while the validation temperature is set to 0.4. The mini-batch size is 64, and the KL-divergence loss coefficient is 0.01. Finally, the discount factor $\gamma$ is set to 0.95.

**Computing Resources.** All online RL experiments were conducted using eight NVIDIA A100 GPUs (40GB each).

## E QUALITATIVE ANALYSIS ON TIPS

### E.1 MORE EXAMPLES OF GENERATED TIPS

Following the example of the generated tips in Section 4.1, below are more detailed examples of how the tips evolve as the task progresses.

---

**ScienceWorld <power-component> task**

You focused on the red light bulb but did not complete the task of turning on the red light bulb. You are in the hallway and need to find a way; At that timestep, The specific action your took was focus on red light bulb. **score -100.0**

Trajectory far from completion; connected battery to red wire but not in correct configuration; gained insights on available objects but missed key steps in circuit creation.; At that timestep, The specific action your took was connect battery cathode to red wire terminal 1. **score 0.0**

Connected green wire to green light bulb but did not find the battery to complete the circuit. Trajectory is incomplete as the task requires powering the green light bulb.; At that timestep, The specific action your took was connect battery to green wire terminal 1, connect green wire terminal 2 to green light bulb, connect red wire terminal 1 to battery. **score 7.0**

Connected green wire to green light bulb, but task not fully completed due to missing battery connection.; At that timestep, The specific action your took was connect green wire to green light bulb. **score 13.0**

Task incomplete, terminal connections incorrect; got red wire, battery, and red light bulb, but terminal connections not successfully made to power red light bulb.; At that timestep, The specific action your took was connect battery cathode to red wire terminal 1. **score 27.0**

I persisted in connecting the green wire to the green light bulb but the circuit was interrupted by other wires, affecting the task completion.; At that timestep, The specific action your took was connect green wire terminal 2 to green light bulb. **score 33.0**

You moved to the workshop, connected the circuit, but the green light bulb is still not powered on due to the lack of a power source and proper connections.; At that timestep, The specific action your took was turn on green light bulb. **score 83.0**

The green light bulb was turned on, but the door to the workshop was closed repeatedly.; At that timestep, The specific action your took was open door to workshop. **score 83.0**

---

---

**ScienceWorld <chemistry-mix-paint-secondary-color> task**

You moved to the greenhouse but did not find the necessary materials to create green paint, indicating the task cannot be completed with the current information and location.; At that timestep, The specific action your took was go to kitchen. **score -100.0**

Failed to execute the task due to unknown actions; collected information about paint colors and mixing process but unable to complete the task of creating green paint.; At that timestep, The specific action your took was pour cup containing yellow paint in art studio in bowl, pour cup containing blue paint in art studio in bowl, mix bowl. **score 0.0**

Failed to find a way to the kitchen or other useful areas; picked up an orange but it's not suitable for making green paint. At that timestep, The specific action your took was open door to kitchen. **score 10.0**

Failed to create green paint; mixed blue and yellow paint but action was ambiguous. Collected info on room layout and objects but not sufficient to complete task.; At that timestep, The specific action your took was open door to workshop. **score 30.0**

The agent attempted to pour yellow paint into the bowl in the art studio but could not find the appropriate action, indicating a need to directly interact with the paint containers; At that timestep, The specific action your took was pour cup containing yellow paint in art studio in bowl and pour cup containing blue paint in art studio in bowl and mix bowl. **score 40.0**

The task is to create green paint by mixing blue and yellow paint, but I repeatedly tried to pour blue paint into the bowl without success.; At that timestep, The specific action your took was pour cup containing blue paint in art studio in bowl. **score 40.0**

Created green paint by mixing yellow and blue paint, but focus action was unnecessary as the paint was already in the bowl. At that timestep, The specific action your took was pour cup containing yellow paint in art studio in bowl. **score 80.0**

---

## E.2 EFFECTS OF TIPS ON EXPLORATION BEHAVIOR

This section provides a qualitative analysis of how tips promote exploration.

---

**ScienceWorld <power-component> task**

**[Without Memory]**

Task : Your task is to turn on the red light bulb. First, focus on the red light bulb. Then, create an electrical circuit that powers it on.
Observation : This room is called the hallway. In it, you see:
- the agent
- a picture
You also see:
- A door to the workshop (that is open)
...
Action : focus on red bulb

**# Task Failed**

**[With Memory-Augmented Prompting]**

Task : Your task is to turn on the red light bulb. First, focus on the red light bulb. Then, create an electrical circuit that powers it on.
Observation : This room is called the hallway. In it, you see:
- the agent
- a picture
You also see:
- A door to the workshop (that is open)
...
tips : Here are some memories you collected in your previous exploration:
Focus on red light bulb but cannot find it in the hallway.; At that timestep, the specific action your took was focus on red light bulb; Eventually you got the score -100.0/100.
Action : go to workshop
Observation : You move to the workshop.
...

---

---

**ScienceWorld <chemistry-mix-paint-secondary-color> task**

| [Without Memory] | [With Memory-Augmented Prompting] |
|---|---|
| Task : Your task is to use chemistry to create green paint. When you are done, focus on the green paint.
Observation : This room is called the hallway. In it, you see:
- the agent
- a picture
You also see:
- A door to the art studio (that is open)
...
Action : pour cup containing yellow paint in hallway in bowl

**# Task Failed** | Task : Your task is to use chemistry to create green paint. When you are done, focus on the green paint.
Observation : This room is called the hallway. In it, you see:
- the agent
- a picture
You also see:
- A door to the art studio (that is open)
...
tips : Here are some memories you collected in your previous exploration:
Need to find a green pigment or mixture to create green paint.; At that timestep, the specific action your took was pour cup containing yellow paint in hallway in bowl; Eventually you got the score -100.0/100.
Need to find a green pigment or mixture to create green paint.; At that timestep, the specific action your took was open door to art studio; Eventually you got the score -100.0/100.
Action : go to art studio
Observation : You move to the art studio.
... |

As shown in the examples above, an agent without memory tends to repeat the same mistakes because it cannot incorporate feedback from previous failures into future attempts. In contrast, with memory-augmented prompting, the agent can refer to its past unsuccessful attempts, use them as guidance, and actively avoid repeating those errors. This enables the agent to explore novel and more effective behaviors, ultimately expanding its search capabilities and boosting learning performance.

## F   MORE ABLATION STUDY

### F.1   MODE SELECTION PROBABILITY

As discussed in Section 4.2, EMPO$^2$ employs a memory-rollout probability $p$ during the rollout phase and an off-policy update probability $q$ during the update phase. We conduct comprehensive ablation studies to systematically investigate the effects of these hyperparameters $p$ and $q$.

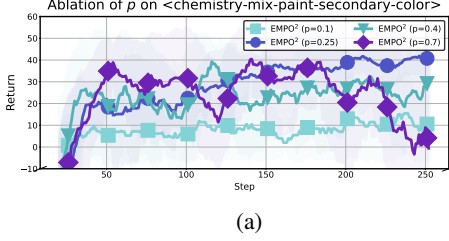
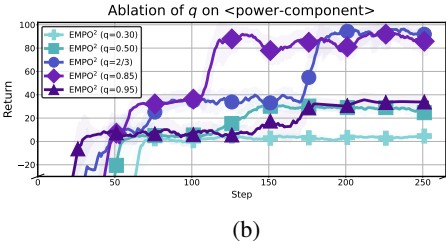

(a)                                        (b)

Figure 10: (a) EMPO² learning curves with varying $p$, (b) with varying $q$

**Ablation on p (Memory Rollout Probability)**: We evaluated $p \in \{0.1, 0.25, 0.4, 0.7\}$ on the chemistry-mix-paint-secondary-color task. When $p = 0.1$, performance degrades significantly because EMPO$^2$ effectively collapses to GRPO, confirming the importance of memory.

Both $p = 0.4$ and $p = 0.7$ show faster initial learning due to more aggressive knowledge internalization, although $p = 0.7$ exhibits some fluctuations in the later stages. Our choice of $p = 0.25$ provides stable convergence across diverse tasks.

**Ablation on q (Off-Policy Update Probability)**: We tested $q \in \{0.3, 0.5, 0.67, 0.85, 0.95\}$ on the `power-component` task. Extreme values ($q = 0.3$ or $q = 0.95$) underperform: very large $q$ overemphasizes distillation at the expense of training the memory policy, while small $q$ slows knowledge internalization. Notably, $q = 0.85$ achieves faster early exploration than our default $q = 2/3$. This aligns with our expectations, as the default hyperparameters prioritize overall robustness rather than task-specific optimization. Therefore, it is natural that more optimal settings exist for particular tasks, highlighting the robustness of EMPO² within a reasonable hyperparameter range.

These results confirm that EMPO² performs effectively across a broad hyperparameter range. Our default settings represent a well-balanced configuration that generalizes across multiple tasks without task-specific tuning, while the algorithm remains adaptable when further optimization is desired.

## F.2 ROLE OF INTRINSIC REWARD

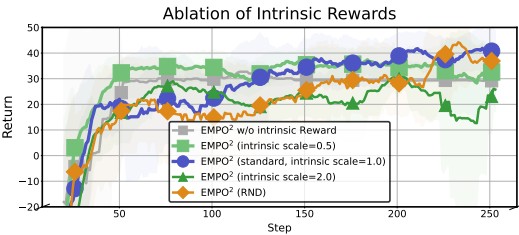

Figure 11: EMPO² learning curves with different intrinsic reward configurations on ScienceWorld `chemistry-mix-paint-secondary-color` task. We compare our full method against four variants: scaling the intrinsic reward coefficient by 0.5× and 2×, substituting it with a Random Network Distillation (RND) bonus, and its complete removal (w/o Intrinsic Reward).

To further investigate the role of the intrinsic reward in our proposed algorithm, EMPO², we conduct an ablation study to examine its impact. We compare our full method against variants with different intrinsic reward coefficients (0.5× and 2×), a complete removal of the intrinsic reward, and its replacement with a standard exploration bonus based on Random Network Distillation (RND) (Burda et al., 2018a). For the RND baseline, we adopt the same hyperparameter configuration as in the original work. The results of these experiments are presented in Figure 11.

Altering the intrinsic reward's scale mainly affects the learning dynamics. A smaller coefficient (0.5×) leads to a smoother but slower convergence, whereas a larger one (2×) introduces minor instabilities. Notably, all variants using an intrinsic reward—including the RND-based one—converge to a similar level of final performance. However, removing the intrinsic reward entirely causes learning to plateau at a lower level, suggesting its necessity in preventing the policy from collapsing into homogeneous behaviors by encouraging sufficient exploration. Overall, these results indicate that EMPO² is robust to the specific mechanism and scale of the intrinsic reward, which primarily influence the stability and speed of learning rather than the final outcome.

# G  ANALYSIS OF COMPUTATIONAL COST

## G.1  COST ANALYSIS OF MEMORY-AUGMENTED ROLLOUTS

We analyzed the additional computational overhead introduced by the memory mechanism in EMPO². During the rollout phase, this mechanism incurs extra costs related to tip generation, retrieval, and storage. For the analysis, we conducted experiments using the Qwen2.5-7B-Instruct model on 8 A100 40GB GPUs.

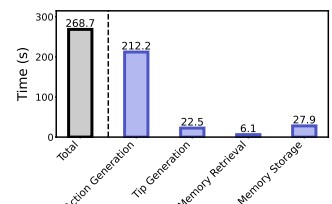

As reported in Figure 12, the memory mechanism adds approximately 50.4 seconds per iteration, which accounts for about 19% of the total rollout time. Among these, tip generation and the subsequent storage of tips in memory account for a substantial portion of the cost. Therefore, while we have verified that the memory mechanism substantially aids exploration and significantly improves learning efficiency, it is more desirable to internalize these

Figure 12: A breakdown of the time each component spends during the rollout of each training step.

benefits within the model parameters themselves rather than relying on the mechanism continuously—both to enhance the model's inherent capabilities and to improve overall efficiency.

## G.2  COST ANALYSIS OF TOTAL TRAINING TIME

Compared to GRPO, the training time of EMPO$^2$ is primarily influenced by two factors:

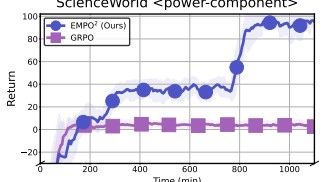

- **The memory component**: As discussed in the previous section, the memory component accounts for 19% of the total rollout time. Since memory-augmented prompting is selected with probability $(1 - p = 0.25)$ (as described in Section 4.2), this implies that, on average, 19% of the rollout time is incurred with a 25% probability.

- **The response length:** In LLM-based RL training, rollout time constitutes a major portion of the total cost. As the response length increases, the rollout itself becomes slower, and the time required for log-probability computation and actor updates in-

Figure 13: Time–performance curves for EMPO$^2$ and GRPO on ScienceWorld `power-component` task.

creases accordingly. In our experiments, we found that the response length of EMPO$^2$ is generally longer than that of GRPO. We attribute this to the model spending more time reasoning and exploring when given the tips, which we believe enhances its exploration behavior and ultimately improves performance.

To ensure a fair comparison with GRPO from a training-time perspective, we plot the performance in Figure 13 using training time on the x-axis. The results show that, even under this perspective, EMPO$^2$ exhibits substantially higher efficiency than GRPO.

# H  THE USE OF LARGE LANGUAGE MODELS

We used a LLM to polish the writing of the manuscript. The LLM was not employed in any aspect of research ideation, experimental design, or analysis.

