# OpenReview forum: "Exploratory Memory-Augmented LLM Agent via Hybrid On- and Off-Policy Optimization"
_ICLR.cc/2026/Conference — ICLR 2026 Poster_

### Official Review · Reviewer_r39R · 2025-10-16

**Soundness:** 3
**Presentation:** 3
**Contribution:** 2
**Rating:** 8
**Confidence:** 3

**Summary:**

This paper introduces a reinforcement learning framework, EMPO², designed to enhance exploration and generalization in large language model (LLM) agents. Existing RL-based LLM agents often overfit to pretrained knowledge and fail to discover novel states. EMPO² addresses this by combining on-policy and off-policy updates with a self-generated memory mechanism—where the agent reflects on past failures to produce “tips” that guide future rollouts.

It internalizes useful behaviors into model parameters while maintaining adaptability via external memory through hybrid optimization. Evaluations on ScienceWorld and WebShop show substantial gains over GRPO with strong out-of-distribution adaptability requiring no parameter updates. The framework is proposed to be a step toward building self-improving, memory-aware LLM agents capable of efficient exploration and continual learning.

**Strengths:**

- Novelty in integration of memory and RL optimization: The paper introduces a hybrid framework (EMPO²) that unifies parametric (on-policy) and non-parametric (off-policy) learning, bridging memory-augmented reasoning and reinforcement learning. This combination is conceptually novel and addresses a long-standing gap between reflection-based and RL-based LLM agents.

- Effective exploration mechanism: By incorporating self-generated reflective memory (“tips”), the method enables autonomous correction of past errors and promotes deeper exploration without additional supervision. It is a meaningful improvement over prior static-memory or fixed-parameter approaches.

- The method demonstrates substantial gains on two challenging multi-step reasoning benchmarks, ScienceWorld and WebShop, with comparably significant improvement over GRPO. It also shows good out-of-distribution generalization with zero-shot adaptability.

- The study includes comprehensive ablation analyses, comparisons across offline, online, and non-parametric baselines, and even computational cost breakdowns, supporting the robustness of the findings.

- The paper is clearly written, with structured algorithmic explanations, detailed pseudocode, and implementation appendices.

- Broader significance: EMPO² provides a promising direction toward self-improving, memory-aware, and generalizable LLM agents, with potential applications in embodied AI, web interaction, and general decision-making systems.

**Weaknesses:**

- The novelty is limited: The core contribution of EMPO² lies in combining existing components, such as memory reflection, on/off-policy RL, and intrinsic rewards, rather than introducing a fundamentally new algorithmic principle or theoretical insight. The innovation is primarily architectural rather than conceptual, which makes its novelty kind of limited.

- The paper does not provide a formal or empirical analysis explaining why the hybrid on/off-policy mechanism stabilizes exploration or improves generalization. Key hyperparameters such as the rollout and update probabilities (p,q) are heuristic, with no clear sensitivity or convergence analysis.

- Evaluation is confined to two benchmarks (ScienceWorld and WebShop), both text-based and reasoning-oriented. The framework’s effectiveness in broader or more complex environments—such as robotics, code synthesis, or multimodal RL—remains untested.

- The study does not analyze the semantic quality of generated “tips” or demonstrate how they concretely guide exploration. Without such analysis, it is unclear whether the model truly learns generalized reasoning strategies or simply memorizes patterns.

- Naming of the Method: the name of the method is kind of confusing at the first time. The "square" symbol is like the footnote, which is ambiguous. Since it is an acronym, it is better to show the full name at the first time of the appearance (e.g. in the abstract, you should show the full name at the first time it appears).

**Questions:**

- Q1: Justification of the hybrid update design: Could the authors elaborate on why combining on-policy and off-policy updates yields more stable or effective exploration in LLM agents? A theoretical or empirical rationale (e.g., ablation across different ratios of on/off-policy updates) would strengthen the methodological foundation.
- Q2: Sensitivity to hyperparameters p and q: The rollout and update probabilities seem chosen heuristically (p=0.75, q=1/3). How sensitive is EMPO² to these settings? Have the authors explored how different sampling ratios affect training stability, exploration depth, or convergence?
- Q3: Role and quality of generated “tips”: The memory mechanism is central to EMPO². Could the authors provide qualitative examples or a deeper analysis of what kinds of tips are most beneficial? Do these tips generalize semantically across tasks, or do they mainly encode task-specific heuristics?

---

> ### Author Response · Authors · 2025-11-19
>
> Thank you for your positive evaluation of our work and for suggestion the ablation experiments and analysis, which has been very helpful in improving our research.
>
> &nbsp;
>
> ### [W1, Q2] Regarding p and q ablation
>
> Thank you for this insightful question. We have added comprehensive ablation studies in **Appendix F.1** to systematically investigate the hyperparameters *p* and *q*.
>
> - **Ablation on p (Memory Rollout Probability):** We evaluated p ∈ {0.3, 0.6, 0.9} on the chemistry-mix-paint-secondary-color task. When p = 0.9, performance degrades significantly as EMPO² essentially reduces to GRPO, confirming the importance of memory. Both p = 0.3 and p = 0.6 show faster initial learning due to more aggressive knowledge internalization, though p = 0.3 exhibits minor late-stage fluctuations. Our choice of p = 0.75 provides stable convergence across diverse tasks.
> - **Ablation on q (Off-Policy Update Probability):** We tested q ∈ {0.05, 0.15, 0.5, 0.70} on the power-component task. We found that extreme values, such as 0.05 or 0.70, led to suboptimal performance. A very small q over-emphasizes distillation at the expense of memory policy training, while a large q slows the knowledge internalization process. Notably, q = 0.15 achieved faster early exploration compared to our default of q = 1/3. This aligns with our expectations, as the default hyperparameters are designed for overall robustness rather than task-specific optimization. Thus, it is natural that a more optimal setting exists for a particular task, underscoring the robustness of EMPO² within a reasonable hyperparameter range.
>
> These results confirm that EMPO² performs effectively across a broad hyperparameter range (p ∈ [0.6, 0.75], q ∈ [0.15, 0.5]). Our default settings represent a well-balanced configuration that **generalizes across multiple tasks without task-specific tuning**, while the algorithm remains adaptable when optimization is desired.
>
> &nbsp;
>
> ### [W2] Regarding more complex environments
>
> We sincerely thank the reviewers for this constructive suggestion. We respectfully note that ScienceWorld and WebShop are already challenging, non-toy benchmarks widely used in LLM agent research. In addition, our ablation studies demonstrate EMPO²'s promising generalizability, as Figure 8 shows strong cross-task adaptation with minimal trials and no parameter updates, which suggests good scalability to more realistic environments. We acknowledge the value of broader evaluation and have outlined future directions in Conclusion (Section 7). We remain committed to exploring more complex real-world interactive environments in future work.
>
> &nbsp;
>
> ### [W3, Q3] Qualitative analysis of how the tips help
>
> We have included the qualitative analysis of the tips in **Appendix E.2** of our revised manuscript. As shown there, the tips provide a summary of the previous rollout along with guidance regarding the actions taken. This guidance helps the agent avoid repeating past failures and enables new action exploration. For example, if the agent previously failed because it did not enter a new room, the tip informs the agent of this, and the agent then takes a new action that leads it toward the new room.
>
> Regarding the last question, the tip contains information about previous attempts of the same task, so it is task-specific.
>
> &nbsp;
>
> ### [W4] Regarding the Method Naming
>
> Thank you for your suggestion. We acknowledge that introducing the abbreviation without first providing the full name may cause confusion. In response, we have revised the manuscript to clarify the full name at the abstract.
>
> &nbsp;
>
> ### [Q1] Justification of the hybrid update design
>
> Thank you for the question. Our hybrid design intentionally leverages the complementary roles of on-policy and off-policy updates to address the stability–efficiency trade-off. On-policy updates ensure stability by constraining the policy to remain close to its current behavior distribution. In contrast, off-policy updates enable the agent to internalize efficiency gains from auxiliary signals such as tips, which we interpret as reward-guided knowledge distillation. Specifically, the agent can perform inference with access to tips and then distills the resulting improved behaviors so that they can be reproduced without relying on them. The distilled policy is subsequently refined through on-policy updates. Alternatively, this process can be viewed as continuously incorporating tips into on-policy learning, thereby preventing convergence to suboptimal solutions without loss of stability. By iteratively alternating between these stages, the agent achieves both stability and efficiency.
>
> Our ablations (Sec. 6.3) offer direct empirical evidence: removing either component results in a sharp performance decline, confirming that their synergy is essential for both stable and efficient exploration.

---

### Official Review · Reviewer_c3bg · 2025-10-22

**Soundness:** 3
**Presentation:** 3
**Contribution:** 4
**Rating:** 8
**Confidence:** 4

**Summary:**

Online reinforcement learning (RL) has recently emerged as a powerful method to improve reasoning and agentic capabilities of large language models (LLMs). However, these methods generally employ on-policy rollouts, and past failed attempts deliver no information other than single scaler reward. For hard tasks where models can be consistently wrong, on-policy samples do not recover any new information, and models may never learn how to solve these tasks via RL.

A possible alternative to this approach is to incorporate memory into LLM agents — LLMs can read their past rollouts, figure out where they had gone wrong/what they could have done differently, and use them to collect future experience. This paper proposes EMPO$^2$, an end-to-end framework where LLMs generate memory in the form of hints from prior rollouts, incorporate them in future rollouts to collect better experience, and uses a mixture of on and off-policy optimization to both improve the model with these hints, and distill the behavior with hints into the model via off-policy learning to retain good behavior on these tasks even when hints are not present. Experiments on multi-turn agentic environments like WebShop and ScienceWorld show superior performance of the proposed method compared to pure on-policy online RL methods like GRPO.

**Strengths:**

**Overall, the paper is quite strong and I recommend acceptance of the paper.**

## Novelty

The paper proposes a mechanism for self-generating memory, incorporating memory into the rollout mechanism in order to avoid past mistakes, promote exploration and achieve better rollouts. Moreover, to the best of my knowledge, this paper is the first to use off-policy learning to then distill back these **hint-augmented** prompts back into the model’s parametric knowledge. This is remarkable and also what I was looking for in the online RL system, Kudos to the authors for making it work so nicely.

To the best of my knowledge, this was only possible in an offline manner using SFT via context distillation [1] (this is an important work that should be cited and discussed however), but no one has done it using online RL before.

The hints are self-generated and do not use a stronger model, which makes the work more appealing.

## Strength of results

The results provided on two benchmarks are very strong, showing remarkable improvement over regular on-policy online RL. This may also unlock improvement in cases where on-policy RL has failed, and have potential implications beyond what the paper presents, i.e., on reasoning tasks like math/coding.

## Memory

The fact that models can generate their own memory and use it in future effectively, **despite not being an entirely novel idea**, is very nice to see in practice.

**Weaknesses:**

As mentioned above, I really like this paper. However, I would note the following weaknesses:

## Comparison on single turn reasoning tasks

The idea of off-policy updates using previously generated hints can be useful beyond the tasks used in this paper. Particularly, this can help regarding single turn reasoning tasks like math/coding.

This is the single most important point where the paper's results can be improved. **If the authors can demonstrate the usefulness of their framework on these tasks, and respond to the other weaknesses/questions I mention below, I am very likely to increase my score on this paper further.**

## Adding a comparison of performance vs GPU hours

The proposed method is inherently more computationally expensive compared to regular on-policy GRPO. **While Appendix E provides a rudimentary analysis of the breakdown of compute spent on various components of the proposed method, no comparison with GRPO is given.** The authors should include a plot with

1. X-axis: GPU hours/flops/some other measure of compute

2. Y-axis: performance

In the main paper, to make the comparison fairer with GRPO.

## No ablation for intrinsic exploration reward

The paper uses an intrinsic exploration reward for novel states, in order to encourage the model to explore unseen/sufficiently novel states. However, I could not find an ablation of the proposed method showing how the performance differs in case this reward is not added/for different choices of the intrinsic exploration reward. To simply put it, it is unclear what the effect of this component of the proposed method is on the performance of the method.

## Experiments using only one base model

All the experiments in the paper are done using only one base model. While the results are strong, it is unclear if the gains come from model specific pretraining/finetuning for Qwen2.5-7B-Instruct. Including results on models from different companies/different pre-training would make the paper significantly stronger.

## No learning component for reward generation

**This is more of an after-thought for future work instead of a serious weakness.** The proposed algorithm does not incentivize better memory generation. However, some proposed memory/hints can be better at steering future generations compared to others, and the model is never incentivized **directly** to generate better memory/hints (it may get incentivized indirectly via reward on memory-augmented rollouts). This needs to be addressed to make the learning better.

(**Minor**)

An important prior work for section 3 discussing LLM agents to be information seeking is Paprika [2]. Similarly, context distillation [1] for learning to behave the same way without hints as whenever hints are available is an important prior work that should be cited and discussed.

**Questions:**

(**Question 1: Advantage Estimation**)

Based on the definition of advantage in line 106-107, am I correct to understand that there is no per-turn advantage? It seems like the advantages are calculated using the entire sum of rewards in each trajectory.

(**Question 2: Calculating Importance Sampling Ratio**)

It is not clear to me how the importance sampling ratios are calculated for off-policy updates. Based on the text and Algorithm 1 from Appendix A, it seems the old log probs ($log \pi_{\theta_{old}}$) are calculated using an off-policy manner (i.e., without generated hints). But how is the current log probs ($log \pi_\theta$) calculated for the off-policy updates? With or without the hints? More generally, could you elaborate how $log \pi_\theta$ is calculated for all different cases (regular on-policy updates, on-policy updates with hints, off-policy updates without hints)? Adding a table clarifying these cases to the main paper would help a lot regarding the clarity of the paper.

(**Question 3: Figure 8**)

I am a bit confused about this figure.

Why are the EMPO and GRPO plots split across two different panels and not on the plot? It is much harder to compare, at least at the first glance, what the performance difference is.
Why does the starting point between EMPO and GRPO vary? Is it because they already went through one round of training with their respective algorithm on the previous task?
What happens if you take the checkpoint resulting from GRPO and run EMPO on top of it, and vice versa (run GRPO on the checkpoint from EMPO) on the new task?

(**Question 4: ScienceWorld**)

What does return/reward mean for ScienceWorld? Is there some other metric like task success rate/completion rate beyond just reward coming from different subgoals/components within a task? Could the authors report that?

(**Question 5: Performance Difference between ScienceWorld and WebShop**)

The proposed method seems to outperform online GRPO quite heavily on ScienceWorld, but not so much on WebShop. Is there a reason/explanation for this?

(**Question 6: Example of how the hints help**)

Could the authors put rollouts with/without hints side-by-side in the appendix, to showcase an example of how the hints help generate better rollouts/avoid common or repeated mistakes?

# References

[1] Learning by Distilling Context, https://arxiv.org/abs/2209.15189

[2] Training a Generally Curious Agent, https://arxiv.org/abs/2502.17543

---

> ### Author Response · Authors · 2025-11-19
>
> Thank you for your positive evaluation of the value and significance of our work and for suggesting meaningful directions to further enhance our work.
>
> &nbsp;
>
> ### [W1] Comparison on single turn reasoning tasks
>
> Thank you for the interesting question. We believe that providing linguistic continuity between rollouts as a form of memory could also yield benefits in single-turn math domains. Due to the time and resource limit, we are still investigation the setting for the single turn experiment. We’re not sure whether we’ll be able to share the results before the rebuttal end. If we do obtain results, we will share them.
>
> &nbsp;
>
> ### [W2] Adding a comparison of performance vs GPU hours
>
> Thank you for the suggestion. We have added this comparison in Appendix F.2 of the revised manuscript. In summary:
>
> - EMPO² requires more training time than GRPO for two main reasons:
>     1. The memory module increases rollout time by about 19%, and this overhead appears when memory-augmented prompting is used (25% probability).
>     2. EMPO² tends to generate longer responses, which increases rollout time and the cost of log-probability computation and model updates, as the model performs more reasoning and exploration when guided by tips.
> - Despite this, EMPO² requires significantly less training time to achieve the same level of performance and yields a notably higher final reward.
>
> &nbsp;
>
> ### [W3] Regarding intrinsic reward ablation
>
> We have added an ablation study on the intrinsic reward in **Appendix F.2** to evaluate its impact and, more importantly, to demonstrate the robustness of our algorithm to its specific design.
>
> Our results confirm that EMPO² is indeed robust to the scale and mechanism of the intrinsic reward. Specifically, while varying the reward coefficient (0.5× and 2×) or substituting our reward with a standard RND-based bonus does affect intermediate learning dynamics (e.g., stability and convergence speed), all these variants ultimately converge to a similar level of final performance.
>
> This shows that although intrinsic reward is necessary, as seen from performance collapse when it is removed, our algorithm is robust to its specific design.
>
> &nbsp;
>
> ### [W4] Regarding the use of the base model
>
> We agree that evaluating our approach on a broader set of models (e.g., Mistral, the Llama family, and larger-scale models) would further strengthen the work and help disentangle model-specific pretraining/finetuning effects. Due to current computational resource constraints, our experiments focus on Qwen2.5-7B-Instruct, and as noted in the Conclusion (Section 7), we expect that larger-scale models will amplify the effects of our self-tip generation and hybrid optimization, and we plan to further explore this as future work.
>
> &nbsp;
>
> ### [W5] No learning component for reward generation
>
> Thank you for suggesting such a valuable future research direction! We also agree that providing additional rewards to encourage more helpful tip (memory) generation could lead to better memory utilization and improved learning efficiency. In the present paper, we deliberately keep the memory component simple and modular to focus on the core mechanism of memory-augmented rollouts, and based on your suggestion, we plan to explore more in-depth methods for jointly reinforcing better tip generation in future work.
>
> &nbsp;
>
> ### [W6] Regarding more related works
>
> Thank you for pointing out these relevant prior works. In the revised version, we added the corresponding citations and included a brief discussion of the method and its relation to our work in Section 5.

---

> > ### Author Response · Authors · 2025-11-19
> >
> > ### [Q1] Regarding advantage estimation
> >
> > Yes, your understanding is correct. In our setup the advantage is defined over the whole trajectory, and GRPO itself does not support per-turn (per-step) advantages.
> >
> > &nbsp;
> >
> > ### [Q2] Regarding calculating importance sampling ratio
> >
> > Thank you for this insightful question. To further clarify this mechanism, we added an explicit explanation and a summary table in the Appendix C of the revised manuscript. The importance ratios $\rho_\theta$ are computed as follows:
> >
> > | Update Mode | Rollout Condition | Current Log Prob | Old Log Prob | Ratio $\rho_\theta$ |
> > | --- | --- | --- | --- | --- |
> > | Regular On-Policy | No tips | $\pi_\theta(a_t \mid s_t,u)$ | $\pi_{\theta_{\text{old}}}(a_t \mid s_t,u)$ | Standard on‑policy (no tips) |
> > | On-Policy w/ Tips | With tips\(_t\) | $\pi_\theta(a_t \mid s_t,u,\text{tips}_t)$ | $\pi_{\theta_{\text{old}}}(a_t \mid s_t,u,\text{tips}_t)$ | Both with tips |
> > | Off-Policy | With tips\(_t\) (rollout) | $\pi_\theta(a_t \mid s_t,u)$ | $\pi_{\theta_{\text{old}}}(a_t \mid s_t,u,\text{tips}_t)$ | Current **without** tips / old **with** tips |
> >
> > Off-policy updates only happen when actions are sampled under $\pi_{\theta_{\text{old}}}(\cdot \mid s_t,u,\text{tips}_t)$ (these are old log-probs *with* tips), and the current log-probs are recomputed *without* tips — this is exactly the mechanism by which we “internalize” the information provided by the tips into the base policy. Otherwise, we conduct on-policy updates, since there is no difference between the rollout policy and the updated policy.
> >
> > &nbsp;
> >
> > ### [Q3] Regarding Figure 8
> >
> > We acknowledge that presenting the figures separately may have caused some confusion. Because the two results use different y-axis scales, we chose to split the plots to make them easier to interpret.
> >
> > In addition, we want to clarify that the Figure 8 results were obtained from checkpoints post-trained with either GRPO or EMPO2. Our goal in this experiment was to examine how performance evolves during **test time** as the model accumulates memory and receives tips from it. So as you mentioned, since each checkpoint was post-trained with a different algorithm on the previous task, their initial performance naturally differs, and we observe that the model trained with EMPO2 starts with higher initial performance.
> >
> > Regarding your last question, we would like to respectfully note that it pertains to a different type of experiment that is not directly addressed in our study. Your suggestion involves continuing training (i.e., cross-training one algorithm on top of another’s checkpoint), whereas our experiments focus on test-time behavior using fixed checkpoints without any additional training. If there are any aspects we may have misunderstood, we would be grateful for your clarification.
> >
> > &nbsp;
> >
> > ### [Q4] Regarding the meaning of ScienceWorld return
> >
> > In our ScienceWorld experiments, the return is defined as the cumulative reward over an episode, calculated as the sum of the rewards associated with the various subgoals or components of a task. These reward settings follow the default ScienceWorld configuration, which we did not modify.
> >
> > &nbsp;
> >
> > ### [Q5] Regarding the performance gap between domains
> >
> > ScienceWorld is a much more challenging exploration environment. In WebShop, each task has only three action types, whereas ScienceWorld involves 25 action types. Moreover, the horizon differs substantially: WebShop tasks typically require at most around 10 timesteps, while ScienceWorld tasks often require around 30. Therefore, our method proves especially effective in this more demanding setting. Importantly, even in WebShop, where the exploration challenge is relatively mild, our approach still induces better exploration than GRPO and consequently achieves better performance.
> >
> > &nbsp;
> >
> > ### [Q6] Qualitative analysis of how the tips help
> >
> > We have included the qualitative analysis of the tips in **Appendix E.2** of our revised manuscript. As shown there, the tips provide a summary of the previous rollout along with guidance regarding the actions taken. This guidance helps the agent avoid repeating past failures and enables new action exploration. For example, if the agent previously failed because it did not enter a new room, the tip informs the agent of this, and the agent then takes a new action that leads it toward the new room.

---

> > > ### Comment · Reviewer_c3bg · 2025-11-19
> > >
> > > Thanks a lot to the authors for their thoughtful rebuttal! I would encourage them include single turn task results and other things they promised in the rebuttal, for the final version of the paper.
> > >
> > > All my concerns have been addressed, and I have increased the score of the paper from 8 to 10.

---

> > > > ### Author Response · Authors · 2025-11-19
> > > >
> > > > Thank you once again to the reviewer for the positive evaluation of our work! As suggested, we are working on single-turn tasks and will try to reflect the results before the deadline. We will also incorporate the other points mentioned in the rebuttal into the final version of the paper.

---

### Official Review · Reviewer_1fWA · 2025-10-29

**Soundness:** 4
**Presentation:** 4
**Contribution:** 4
**Rating:** 6
**Confidence:** 3

**Summary:**

This paper introduces a memory-augmented RL algorithm for learning effective LLM based policies. The key idea is to use a memory buffer in sampling (rollouts) and policy learning by using the memory to enable more effective exploration in the rollout phase (by being able to reason about past experiences) and using the memory to enable learning a more generalizable policy by performing a combination of on-policy and off-policy (where main policy is not conditioned on memory) learning. Results demonstrate significant improvement in ScienceWorld and moderate improvement in WebShop benchmarks.

**Strengths:**

- Paper is well-motivated and well-written. Justification for improved exploration in RL for LLMs is sound.
- Use of memory in both rollout and update phase is simple yet novel in the context of RL for LLMs.
- Strong results on ScienceWorld which demonstrate the OOD generalization of their method (due to generality of memory).

**Weaknesses:**

- Lack of ablations. The method introduces additional hyperparameters and components, the effects of which are largely undocumented.
    - Effect of intrinsic reward component. What is the effect of this component on the performance of the final policy (paper only documents the effect on policy entropy)? How generalizable is this reward term? It seems as if it may require further reward-shaping (i.e. tuning similarity threshold) to generalize to newer domains where naive state similarity may lead to poor performing rollouts (e.g. see static noise TV example from Pathak et al).
    - Effect of sampling proportion ($p$) between memory-free and memory-augmented rollouts. What is the effect of varying $p$?
    - Effect of update proportion ($q$) between on- and off-policy updates. What is the effect of varying $q$?
    - The authors set the KL coefficient to 0.0; does the final model lose its broad generality (e.g. on standard LLM benchmarks)?

[1] Pathak, Deepak, et al. "Curiosity-driven exploration by self-supervised prediction." International conference on machine learning. PMLR, 2017.

**Questions:**

Please see questions listed in weaknesses.

---

> ### Author Response · Authors · 2025-11-19
>
> Thank you for your thoughtful and encouraging feedback, and for the valuable suggestions to further enhance our paper through additional ablations.
>
> &nbsp;
>
> ### [W1] Regarding intrinsic reward ablation
>
> We have added an ablation study on the intrinsic reward in **Appendix F.2** to evaluate its impact and, more importantly, to demonstrate the robustness of our algorithm to its specific design.
>
> Our results confirm that EMPO² is indeed robust to the scale and mechanism of the intrinsic reward. Specifically, while varying the reward coefficient (0.5× and 2×) or substituting our reward with a standard RND-based bonus does affect intermediate learning dynamics (e.g., stability and convergence speed), all these variants ultimately converge to a similar level of final performance.
>
> This shows that although intrinsic reward is necessary, as seen from performance collapse when it is removed, our algorithm is robust to its specific design.
>
> &nbsp;
>
> ### [W2] Regarding p and q ablation
>
> Thank you for this insightful question. We have added comprehensive ablation studies in **Appendix F.1** to systematically investigate the hyperparameters *p* and *q*.
>
> - **Ablation on p (Memory Rollout Probability):** We evaluated p ∈ {0.3, 0.6, 0.9} on the chemistry-mix-paint-secondary-color task. When p = 0.9, performance degrades significantly as EMPO² essentially reduces to GRPO, confirming the importance of memory. Both p = 0.3 and p = 0.6 show faster initial learning due to more aggressive knowledge internalization, though p = 0.3 exhibits minor late-stage fluctuations. Our choice of p = 0.75 provides stable convergence across diverse tasks.
> - **Ablation on q (Off-Policy Update Probability):** We tested q ∈ {0.05, 0.15, 0.5, 0.70} on the power-component task. We found that extreme values, such as 0.05 or 0.70, led to suboptimal performance. A very small q over-emphasizes distillation at the expense of memory policy training, while a large q slows the knowledge internalization process. Notably, q = 0.15 achieved faster early exploration compared to our default of q = 1/3. This aligns with our expectations, as the default hyperparameters are designed for overall robustness rather than task-specific optimization. Thus, it is natural that a more optimal setting exists for a particular task, underscoring the robustness of EMPO² within a reasonable hyperparameter range.
>
> These results confirm that EMPO² performs effectively across a broad hyperparameter range (p ∈ [0.6, 0.75], q ∈ [0.15, 0.5]). Our default settings represent a well-balanced configuration that **generalizes across multiple tasks without task-specific tuning**, while the algorithm remains adaptable when optimization is desired.
>
> &nbsp;
>
> ### [W3] Loss of Generality with KL Coefficient 0
>
> Thank you for raising this important point. Conceptually, any additional fine-tuning (including RL-style optimization) optimizes the model in expectation over a particular training distribution. Therefore, some shift away from the original behavior is indeed unavoidable, even with the KL coefficient. An interesting analogy is that if someone lives in another city for a long time, they might lose the ability to speak their hometown dialect. A more systematic evaluation of this trade-off between task specialization and broad generality is an important direction that we plan to explore in future work.

---

### Official Review · Reviewer_JKue · 2025-11-03

**Soundness:** 2
**Presentation:** 4
**Contribution:** 3
**Rating:** 6
**Confidence:** 2

**Summary:**

The paper introduces the EMPO2 algorithm, which works as follows
- agents roll out trajectories
- at the end they write "tips" for solving the environment
- in future runs, some fraction of the time the agent conditions on hints form past rollouts
- we train on both types of rollouts, including a 3rd variant where the tips are stripped out of the rollout.
- they show improvements above prior works on ScienceWorld and WebShop.

**Strengths:**

Although each individual part of the proposed method is not original, combining them all together under one framework is an important contribution and has not been done before, particularly in the important but still early field of RL on LLMs.

The quality is good, mainly focusing on showing saturation of two in-distribution benchmarks. The paper also demonstrated signs of life on out of distribution benchmarks. The paper also sought to understand each component’s importance by doing ablations, as well as proposed future interesting extensions to the work, such as a similarity-based bonus for novel contributions to the memory bank.

I appreciated the clarity of the communication, the plots, figures, and charts all are well-designed and get the most important points across.

The significance is important, since the standard of RL for LLMs (GRPO) mainly focuses on parametric updates for each training batch rather than encouraging exploration or learning over time across training or episodes.

**Weaknesses:**

I am confused about the hyperparameter choices for choosing to sample between memory and non memory for rollouts and on-policy and off-policy for updates. There isn’t explanation for these choices (½ and ⅓ respectively), and there are no ablations or sweeps (although 6.3 does ablate the entire components).

Some of the plots could have better reporting. For example, figure 1 B not having error bars across the seeds, or figure 8.

For some of the baselines, I am concerned about the reported numbers being derived from other papers . For example, for WebShop, the Naive, Reflexion, GRPO, and GiGPO are taken from the paper. The paper states that all of the hyperparameters are the same for the training methods, which absolves most of my concerns, but RL methods are notorious for subtle implementation differences in the algorithm or environment which may not even be highlighted in the paper making big difference.

I also think scaling this to more “non-toy” training and evaluations would improve the paper’s significance (although not necessary since this is proposing a new method).

**Questions:**

- The off-policy updates seem like they could introduce stability (as you mentioned). Did you consider either using a real off-policy algorithm or introducing an importance-sampling correction where the numerator is the prob with no tips and the denom is prob with tips?

- The off-policy stabilization approach in Fig 6 is interesting. Would be nice to see error bars or multiple seeds there (to make sure it's not that one seed got unlucky in Fig 6).

- A bit ambiguous to me what numbers are reported in Table 1 (and other results) for EMPO2 -- which rollout mode were used to create these? If the "with tips" one, how many rollouts were used to create the memory bank? Same question about baselines that use multiple episodes.

- How were the hyperparameters chosen for sampling from on or off policy?

- Why are there no error bars in table 2 for naive or reflexion?

- Why are there no error bars in figure 8?

---

> ### Author Response · Authors · 2025-11-19
>
> Thank you for your positive feedback about the significance of our work and for the valuable suggestions that improve the comprehensiveness of our results.
>
> &nbsp;
>
> ### [W1, Q4] Regarding p and q ablation
>
> Thank you for this insightful question. Initially, we conducted preliminary hyperparameter tuning and found EMPO² to be robust to variations within a reasonable range. We therefore adopted an intuitive configuration. Now, we have added comprehensive ablation studies in **Appendix F.1** to systematically investigate the hyperparameters *p* and *q*.
>
> - **Ablation on p (Memory Rollout Probability):** We evaluated p ∈ {0.3, 0.6, 0.9} on the chemistry-mix-paint-secondary-color task. When p = 0.9, performance degrades significantly as EMPO² essentially reduces to GRPO, confirming the importance of memory. Both p = 0.3 and p = 0.6 show faster initial learning due to more aggressive knowledge internalization, though p = 0.3 exhibits minor late-stage fluctuations. Our choice of p = 0.75 provides stable convergence across diverse tasks.
> - **Ablation on q (Off-Policy Update Probability):** We tested q ∈ {0.05, 0.15, 0.5, 0.70} on the power-component task. We found that extreme values, such as 0.05 or 0.70, led to suboptimal performance. A very small q over-emphasizes distillation at the expense of memory policy training, while a large q slows the knowledge internalization process. Notably, q = 0.15 achieved faster early exploration compared to our default of q = 1/3. This aligns with our expectations, as the default hyperparameters are designed for overall robustness rather than task-specific optimization. Thus, it is natural that a more optimal setting exists for a particular task, underscoring the robustness of EMPO² within a reasonable hyperparameter range.
>
> These results confirm that EMPO² performs effectively across a broad hyperparameter range (p ∈ [0.6, 0.75], q ∈ [0.15, 0.5]). Our default settings represent a well-balanced configuration that **generalizes across multiple tasks without task-specific tuning**, while the algorithm remains adaptable when optimization is desired.
>
> &nbsp;
>
> ### [W2, Q5, Q6] Regarding error bars
>
> Thank you for your questions, and to make our reporting more comprehensive, we have added the missing standard deviations to Fig. 1B, and Fig. 8 as suggested. For the naive and Reflexion results in Table 2, the prior work that originally reported these scores [Ref.1] did not provide standard deviations, so we were unable to include them.
>
> [Ref.1] Feng, Lang, et al. "Group-in-group policy optimization for llm agent training." NeurIPS 2025.
>
> &nbsp;
>
> ### [W3] Regarding scores from other papers
>
> Thanks for raising this concern. We do agree with you that RL methods are notorious for subtle implementation differences. This is the reason that we want to use results in previous papers rather than re-run all methods, minimizing implementation discrepancies. Reusig is also reasonable here, because WebShop and ScienceWorld are well-established benchmarks with standardized procedures, which should be followed by every paper using these benchmarks.
>
> Based on this principle of reusing baseline results, we copy most of the results from previous work. And we re-ran Retrospex using the official codebase, because Retrospex paper does not contain results obtained with the same setting (base model and test set) as us. So we re-run this method with tuned its hyperparams as well (see Appendix C.1).
>
> &nbsp;
>
> ### [W4] Regarding more complex environments
>
> We sincerely thank the reviewers for this constructive suggestion. We respectfully note that ScienceWorld and WebShop are already challenging, non-toy benchmarks widely used in LLM agent research. In addition, our ablation studies demonstrate EMPO²'s promising generalizability, as Figure 8 shows strong cross-task adaptation with minimal trials and no parameter updates, which suggests good scalability to more realistic environments. We acknowledge the value of broader evaluation and have outlined future directions in Conclusion (Section 7). We remain committed to exploring more complex real-world interactive environments in future work.

---

> > ### Author Response · Authors · 2025-11-19
> >
> > ### [Q1] Regarding other off-policy techniques or algorithms
> >
> > Thank you for your insightful comments regarding off-policy updates. You raise a very valid concern about potential instability, and we appreciate you pointing that out.
> >
> > In fact, our current implementation already matches your second suggestion. As outlined in Section 4.2, we use an importance-sampling correction, where the ratio of the no-tip policy probability to the with-tip policy probability adjusts for the differences. In Appendix C of the revised paper, we have added a detailed explanation of this part.
> >
> > While we didn’t explore a fully off-policy algorithm in this study, we agree that it is a promising avenue for future research, and we also added this part to the conclusion section of the revised paper.
> >
> > &nbsp;
> >
> > ### [Q2] Regarding Fig 6
> >
> > Thank you for your suggestion. We reran the same experiments with two additional random seeds and plotted the mean and standard deviation across three seeds. Although there were slight variations among the three seeds, we were able to confirm once again that our method contributes to overall training stability.
> >
> > &nbsp;
> >
> > ### [Q3] How is EMPO2 evaluated in Table 1
> >
> > The revised paper clarifies that EMPO² models are evaluated without hints, whereas Reflexion models receive memory hints. In both cases, the memory buffer length is set to 10, consistent with EMPO² training.

---

### Author Response · Authors · 2025-11-19
**Summary of Paper Revision**

We would like to express our sincere gratitude to all reviewers for their valuable time and effort in helping improve our work. Below, we summarize the revisions made to the paper.

&nbsp;

### Summary of Paper Revision

Modified sections

- [Abstract] In response to reviewer r39R’s suggestion, we added the full name of our method to the abstract.
- [Section 5] In response to reviewer c3bg’s suggestion, we added relevant related works.
- [Section 6] In response to reviewer JKue’s question, we clarified that the results of Tables 1 and 2 for EMPO² were evaluated without memory at test time.
- [Section 7]  In response to reviewer JKue’s question, we present the exploration of fully off-policy algorithms as a future direction.

New sections

- [Appendix C] Detailed Explanation of Importance Sampling Ratios in Policy Updates
- [Appendix E.2]  Effects of Tips on Exploration Behavior
- [Appendix F] Ablation study on Mode Selection Probability (p and q) and Intrinsic Reward
- [Appendix G.2] Cost Analysis of Total Training Time

---

### Meta-Review · Area_Chair_EAg2 · 2026-01-07

**Summary:**

Reviewers praised the novel integration of memory and hybrid RL to solve LLM exploration. However, I have some critique highlights that the core mechanism's theoretical grounding is light, the random update mixing seems heuristic, and the experiment results might depend on the base model.

**Reviewer Concerns:**

Addressed: Hyperparameter ablations (p, q, intrinsic reward) now show robustness. Qualitative tip examples illustrate the exploration mechanism.
Outstanding (new): A deeper explanation of why the memory-prompting leads to better exploration (beyond example-based claims) is still missing. The random update schedule, while shown to be robust, remains a heuristic choice without theoretical justification. All results use Qwen2.5-7B-Instruct. Gains could be partly due to model-specific biases.

**Reviewer Scores:**

all reviewers gave positive scores

---

### Decision · Program_Chairs · 2026-01-26

Accept (Poster)